# Understanding Molecular Mechanisms of Seed Dormancy for Improved Germination in Traditional Leafy Vegetables: An Overview

**Fernand S. Sohindji, Dêêdi E. O. Sogbohossou, Herbaud P. F. Zohoungbogbo, Carlos A. Houdegbe and Enoch G. Achigan-Dako ***

Laboratory of Genetics, Horticulture and Seed Science, Faculty of Agronomic Sciences, University of Abomey-Calavi, 01 BP 526 Tri Postal, Cotonou, Benin; sohindjisilverfer@gmail.com (F.S.S.); deedik@gmail.com (D.E.O.S.); phanuelherbaud@gmail.com (H.P.F.Z.); houdariscarl@gmail.com (C.A.H.)
* Correspondence: enoch.achigandako@uac.bj; Tel.: +229-95-39-32-83

**Abstract:** Loss of seed viability, poor and delayed germination, and inaccessibility to high-quality seeds are key bottlenecks limiting all-year-round production of African traditional leafy vegetables (TLVs). Poor quality seeds are the result of several factors including harvest time, storage, and conservation conditions, and seed dormancy. While other factors can be easily controlled, breaking seed dormancy requires thorough knowledge of the seed intrinsic nature and physiology. Here, we synthesized the scattered knowledge on seed dormancy constraints in TLVs, highlighted seed dormancy regulation factors, and developed a conceptual approach for molecular genetic analysis of seed dormancy in TLVs. Several hormones, proteins, changes in chromatin structures, ribosomes, and quantitative trait loci (QTL) are involved in seed dormancy regulation. However, the bulk of knowledge was based on cereals and *Arabidopsis* and there is little awareness about seed dormancy facts and mechanisms in TLVs. To successfully decipher seed dormancy in TLVs, we used *Gynandropsis gynandra* to illustrate possible research avenues and highlighted the potential of this species as a model plant for seed dormancy analysis. This will serve as a guideline to provide prospective producers with high-quality seeds.

**Keywords:** seed dormancy; seed germination; molecular biology; genetics; traditional leafy vegetables; *Gynandropsis gynandra*

## 1. Introduction

More than 1000 species were recorded to be used by African rural communities for dietary diversity, medicine purpose, food traditions, and cultural identity [1–4]. Given the potential of African traditional leafy vegetables (TLVs) to cope with varying climate constraints and feed Africa, the production and consumption of the same have been promoted in the continent for the last two decades. For example, the antioxidant system of *Cleome spinosa* Jacq. and *Gynandropsis gynandra* L. (Briq) variously copes with reactive oxygen species (ROS) formation under drought conditions limiting damage to cell structures, lipids, proteins, carbohydrates, nucleic acids, and cell death [5]. Luoh et al. [6] reported also the lowest losses of nutrients in amaranth species and the African nightshade species when cultivating under deficient conditions. Few species such as *Amaranthus* spp., *Solanum scabrum* Mill., and *Solanum macrocarpon* L. have been well domesticated, whereas *Bidens pilosa* L., *Brassica carinata* A. Braun., *Gynandropsis gynandra*, *Corchorus* spp., *Launaea taraxacifolia* Willd., *Talinum triangulare* Willd., etc. are widely used over the continent, but still semi-domesticated species and can grow spontaneously and/or cultivated according to the sociolinguistic groups or regions [3]. The successful promotion of those

crops requires a proper investigation of all aspects related to their life cycle. The first step toward such promotion is to focus on farmers' traits of interest and constraints during cultivation and marketing. Once farmers' priorities are clearly defined, a subsequent step is to investigate crop physiology in relation to the traits of interest for farmers. For several species including *Gynandropsis gynandra*, *Solanum nigrum* L., *Amaranthus* spp., and *Corchorus* spp. the lack of high-quality seeds is the main constraint for full domestication [7]. In fact, seed germination and seed dormancy processes are still unclear for many traditional leafy vegetables. Farmers are confronted with the loss of seed viability, poor and delayed germination of seeds, and inaccessibility to quality seeds limiting all-year-round production of those crops [8,9]. In addition, Adebooye et al. [10] and Sogbohossou et al. [11] brought to light several gaps of knowledge for the improvement of TLVs. They listed, for instance: (1) the lack of extensive germplasm collections; (2) the need to understand the genetic control of key traits; (3) the need to develop and evaluate new cultivars, assess end-users' preferences, and perform multi-environment experiments; (4) the demand for appropriate technical packaging; and (5) the call for sustained efforts for value chain development. However, to reach active domestication, it is important to ensure quality seed availability and find solutions for seed dormancy issues that limit TLVs adoption and production.

　　Seed dormancy is a state of a viable seed, expressed by the inhibition of germination under favourable environmental conditions required for adequate germination [12,13]. It is an adaptive trait that optimizes the distribution of germination over time in a population of seeds [14]. On the other hand, germination is usually related to radicle protrusion, which is normally the visible result of germination [15]. Before this visible aspect, there are many events that begin with the uptake of water by mature dry seed and imbibition, and end with the embryonic axis elongation [16]. Seed dormancy and the absence of favourable environmental conditions for germination result in the absence of germination [12]. Non-germination due to unfavourable conditions is referred to as "quiescence" and enables seed survival for further seedling development under adverse conditions [17]. Seed dormancy appears as a complex quantitative trait under the influence of several genetic, hormonal, physiological, and environmental factors [18]. The pre-harvest sprouting and the absence or Delay of Germination after-ripening are the two undesirable contrasting levels of seed dormancy. Consequently, the constraint about dormancy is twofold: either it is not present in seeds (zero level) leading to pre-harvest sprouting especially for cereals or it is present at high level leading to the absence or Delay of Germination at the desired time [19,20]. There are two types of seed dormancy based on its times of expression: "primary dormancy" developed on the mother plant [21], and "secondary dormancy" induced in previously non-dormant seeds or re-induced in seeds that have lost primary dormancy due to the unfavourable environment factors for germination after seed dispersal [22]. Baskin and Baskin [12] reported various classes of dormancy based on embryo growth potential, seed coat, and seed physiological responses to temperature. Some of the classes were: physical dormancy (PY), physiological dormancy (PD), morphological dormancy (MD), morpho-physiological dormancy (MPD), and combinational dormancy (physical and physiological dormancy).

　　As stated by Koornneef et al. [23], the challenge is to master the initiation and suppression of germination ability through gene identification based on changes in seed transcriptome, proteome, and hormones under different environmental conditions. So far, the majority of molecular genetic studies on seed dormancy were conducted in the model plant species *Arabidopsis thaliana* L. [24] and economically important crops such as *Oryza sativa* L. [25], *Triticum aestivum* L. [26], and *Lycopersicum esculentum* Mill. [27]. Such studies revealed that seed germinability and subsequent seedling development are controlled by two sets of factors: internal factors including proteins, plant hormones (Abscissic–Gibberellic acids balance), chromatin-related factors (methylation, acetylation, histone ubiquitination), related genes (maturating genes and hormonal and epigenetics-regulating genes), non-enzymatic processes, seed morphological and structural components (endosperm, pericarp, seed coat), and external factors, such as light, temperature, salinity, acidity, soil nitrate [28,29]. These biotic and abiotic factors interact and determine the presence or absence of dormancy during seed development, in imbibed mature seeds and in dry seeds. Despite the fact that efforts of plant

biologists, crop geneticists, breeders, and food scientists to understand seed dormancy phenomenon have shed light on physiology, genetic and molecular aspects of seed dormancy, little is known about TLVs species, although seed dormancy is still a challenge in these species as above described.

In this paper, we provide a synthesis of the current state of knowledge about seed dormancy in TLVs and seed dormancy control in plants. We further discussed how to transfer such knowledge to leafy vegetables in tropical areas. We highlighted the case of spider plant (*Gynandropsis gynandra*), a traditional leafy vegetable closely related to the model species *A. thaliana*. This review addressed the following questions: what is the germination ability of the well consumed TLVs? How do we identify seed dormancy regulatory genes in TLVs? How do we study the effects of those genes on seed germination? What approach do we implement to develop new cultivars with higher seed quality and attributes?

## 2. Methods

Literature search was conducted in PubMed Central and Google Scholar databases. Well-studied crops such as *Arabidopsis thaliana*, *Oryza sativa*, *Zea mays* L., *Triticum aestivum*, *Hordeum vulgare* L., *Lycopersicum esculentum*, *Avena sativa* L., *Nicotiana plumbaginifolia* Viv., and African leafy vegetables such as *Gynandropsis gynandra*, *Solanum nigrum*, *Corchorus olitorius* L., *Talinum triangulare* (Jacq.) Willd., and *Amaranthus* spp. (Figure 1) were considered in the search. The keywords used to collect the documents included "seed dormancy", "molecular control", "hormonal control and genetic variation", "germination of African/indigenous/traditional leafy vegetables", and "seed constraints for African/indigenous/traditional leafy vegetables". No date coverage was specified during the search. The relevance of documents was assessed based on the molecular pathways, genetic control, and environmental factors reported about seed dormancy or seed germination. The process consisted of a preliminary screening of titles and abstracts of 243 papers. The reading of full texts helped select 194 relevant papers for further review.

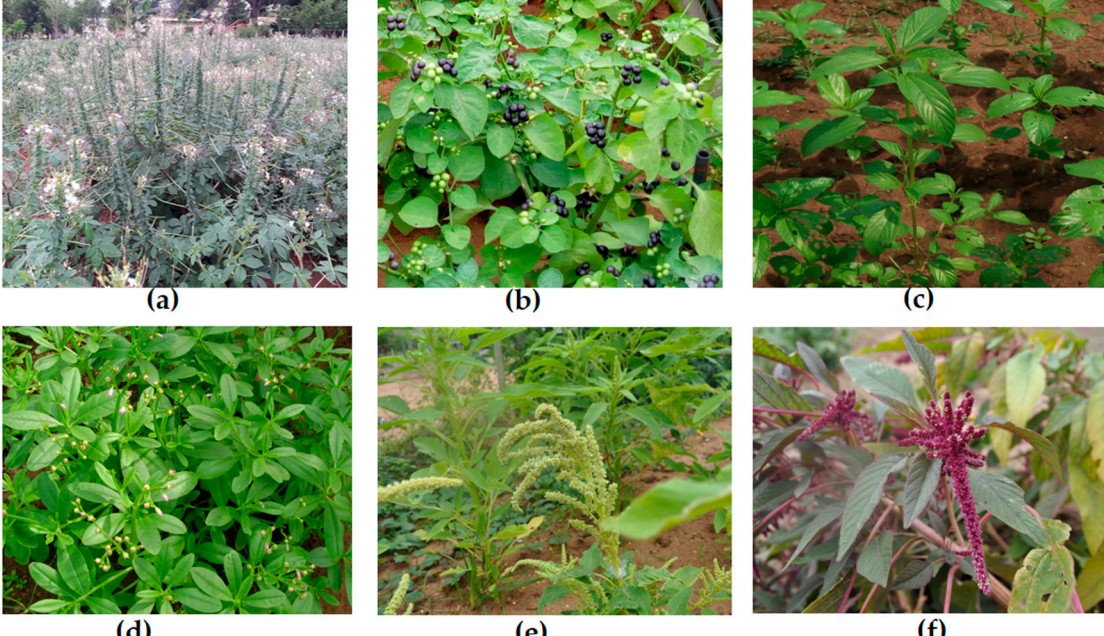

**Figure 1.** Some of important African traditional leafy vegetables widely used across Africa: (**a**) *Gynandropsis gynandra*, (**b**) *Solanum nigrum*, (**c**) *Corchorus olitorius*, (**d**) *Talinum triangulare*, (**e**,**f**) *Amaranthus* spp. Pictures: courtesy of F.S. Sohindji, H.F.P. Zohoungbogbo and E.G. Achigan-Dako.

## 3. Seed Dormancy in Traditional Leafy Vegetables

Seed dormancy prevents seeds to germinate under unfavourable conditions for further growth and development of the plant. In many leafy vegetables, it appears as a challenge while rapid and uniform germination are expected after sowing. So far, physical and physiological dormancy are the main dormancy cases reported in TLVs species from mature freshly harvested seeds to dry seeds [30]. An overview of studies on seed dormancy in TLVs is presented in Table 1.

**Table 1.** Seed constraints and dormancy types reported for traditional leafy vegetables.

| Common Name | Scientific Name (Family) | Seed Constraints | References |
|---|---|---|---|
| Spider plant | *Gynandropsis gynandra* (Cleomaceae) | Primary non-deep physiological dormancy<br>Physical dormancy and secondary dormancy<br>Oxygen barrier between embryo and tissue<br>Low germination of freshly harvested seeds<br>Delayed, poor, and absence of germination<br>Inaccessibility of quality seed for seed analysts and gene bank curators<br>Low vigor and reduced number of viable seeds harvested by farmers<br>Physiological dormancy | [31,32] |
| Jute mallow | *Corchorus olitorius* (Malvaceae) | Loss of viability and poor germination of fresh and old seeds Impermeable seed coat | [8,33] |
| African Nightshade | *Solanum nigrum* (Solanaceae) | Poor germination of seeds<br>Improper seed extraction<br>Deeper level of primary dormancy | [8,34] |
| Waterleaf | *Talinum triangulare* (Portulacaceae) | Dormancy due to the nature of the seed testa<br>Undetermined physiological factors | [35] |
| Amaranths species | *Amaranthus* spp. (Amaranthaceae) | Primary dormancy and secondary dormancy occur among amaranths species | [36] |

Seeds of waterleaf (*Talinum triangulare*) are known to exhibit a kind of dormancy due to the impermeability of the seed testa and some undetermined physiological factors [35]. These authors reported that scarification, alternating temperatures (6–10 °C and 28–35 °C), and constant temperature (34 °C) should enhance germination of waterleaf seeds. Taab and Andersson [34] reported a deeper level of primary dormancy for nightshade (*Solanum nigrum*). Dormancy-breaking treatments with stratification, potassium nitrate, and gibberellic acid failed to show encouraging results, and are often not applicable at farmers' level [34,37]. A loss of viability and poor germination of fresh and old seeds in jute mallow (*Corchorus olitorius*) are associated with the impermeability of jute mallow seed coat [8,33]. In the case of spider plant (*Gynandropsis gynandra*), its cultivation is limited by the fact that its seeds can exhibit a high dormancy lasting for several months. Geneve [38] reported that a primary non-deep physiological dormancy occurs in spider plant while Ochuodho and Modi [39] suspected physical dormancy and secondary dormancy. Ekpong [40] clarified that spider plant seeds are permeable to water but this water is trapped in the tissue between the embryo and the seed coat creating an oxygen barrier. Baskin and Baskin [32] concluded that spider plant exhibits a physiological dormancy. Recently, Shilla et al. [31] reported that there were no dormancy cases on fresh seeds of spider plant according to World Vegetable Center preliminary results. Nevertheless, the seeds can stay in a dormant state for several months before germination is activated and improved with dry storage periods [41–43]. Various levels of seed dormancy such as primary and secondary dormancy occur among amaranths species. For instance, there are primary dormant *Amaranthus retroflexus* L., secondary dormant *Amaranthus paniculatus* L., and non-dormant *Amaranthus caudatus* L. seeds [36]. Seed treatments such as seed holding in low temperature, pre-chilling, and the application of ethylene induce dormancy-breaking and accelerate the germination process in *Amaranthus* seeds [44].

While other factors including harvest time, storage, and conservation conditions can be easily controlled by farmers, breaking seed dormancy requires a thorough knowledge of the seed intrinsic nature and physiology. Farmers' efforts to break seed dormancy are therefore still insufficient to assure

50% to 100% germination. Many seed pre-treatment techniques (Table 2) such as light/dark, cold/warm, tap/distilled water, and physical/chemical scarification have been tested by researchers to break the seed dormancy in crops [13]. Unfortunately, results obtained can be negatively affected by factors such as seed provenance, seasons of production, storage containers, storage period, and storage temperature, to list a few environmental conditions [43]. Therefore, traditional leafy vegetable seeds management is still traditional and farmers, seed companies, gene banks, and researchers require adequate methods for breaking seed dormancy of those species.

**Table 2.** Various strategies for seed dormancy-breaking in African leafy vegetables (*Gynandropsis gynandra*, *Amaranthus* spp., *Corchorus olitorius*, *Talinum triangulare*, *Solanum nigrum*).

| TLVs Species | Strategies for Seed Dormancy-Breaking |
|---|---|
| *Gynandropsis gynandra* | - Stratification for two weeks at 5 °C<br>- 12 h of seed soaking<br>- Scarification with 1000 μM $KNO_3$, 1000 μM $K_2SO_4$, 1000 $(NH_4)_2SO_4$<br>- $GA_3$ application at a concentration of 500 ppm<br>- 60 min of seed pre-washing in running water<br>- 1–5 days of seed pre-heating at 40 °C<br>- Seed dried to 5% moisture content and stored at −20 °C<br>- 3–6 months of after-ripening<br>- Seed puncturing at the radicle end<br>- Darkness and either alternating 20–30 °C or continuously at 30 °C for germination<br>- 1 day of moist chilling at 5 °C |
| *Amaranthus* species | - Holding seeds for 18 months at 6 °C<br>- Pre-chilling treatment<br>- Scarification with 1000 μM $KNO_3$, 1000 μM $K_2SO_4$, 1000 $(NH_4)_2SO_4$<br>- 100 μM $GA_3$ application<br>- Immersion in 2% $KNO_3$ solution for 24 h<br>- Cold seed stratification at 5 °C for 12 days<br>- Germination conditions under light<br>- Application of ethylene, ethephon or 1-aminocyclopropane-1-carboxylic acid |
| *Corchorus olitorius* | - Mechanical scarification<br>- Leaching treatment<br>- Soaking in boiling water for 5 min |
| *Talinum triangulare* | - Activated carbon<br>- Scarification<br>- 5% thiourea treatment<br>- Constantly high temperature (34 °C) for germination<br>- Immersion in 0.2% potassium nitrate solution (24 h)<br>- Immersion in water (24 h) |
| *Solanum nigrum* | - Soaking in GA3 with concentration of 0, 25, 50, 100, 200, and 400 ppm<br>- Wet and dry pre-chilling for 15, 30, and 45 days in 4 °C<br>- Seed coat chemical scarification for 1–3 min<br>- Exposing seed to UV-C radiation for 30 min |

## 4. Seed Dormancy Regulation in Plants

Seed dormancy is regulated by genotypic (internal regulation) and environmental factors during three stages in the persistent soil seed bank such as seed development, after-ripening, and seed germination. [45]. During seed development, some reserves are accumulated in seeds (reserve accumulation). During after-ripening, seeds especially orthodox seeds have ability to survive desiccation (desiccation tolerance). The seed germination stands for mobilization of reserves under favourable condition (reserve mobilization). The viable, mature, and freshly harvested seed can be dormant (primary dormancy) or non-dormant (able to germinate). The favourable conditions during after-ripening lead to the release of primary dormancy seed which becomes non-dormant. When conditions are unfavourable, non-dormant seeds, even those for which primary dormancy

was released, enter the quiescent state before entering into secondary dormancy when unfavourable conditions persist or could be able to germinate under favourable conditions.

The internal regulation of seed dormancy occurs in two main pathways acting in interaction with the environment. There is the hormone-level pathway (indirect pathway) and gene-level pathway (direct pathway). The different known regulators involved in seed dormancy control and their interactions are presented in Figure 2.

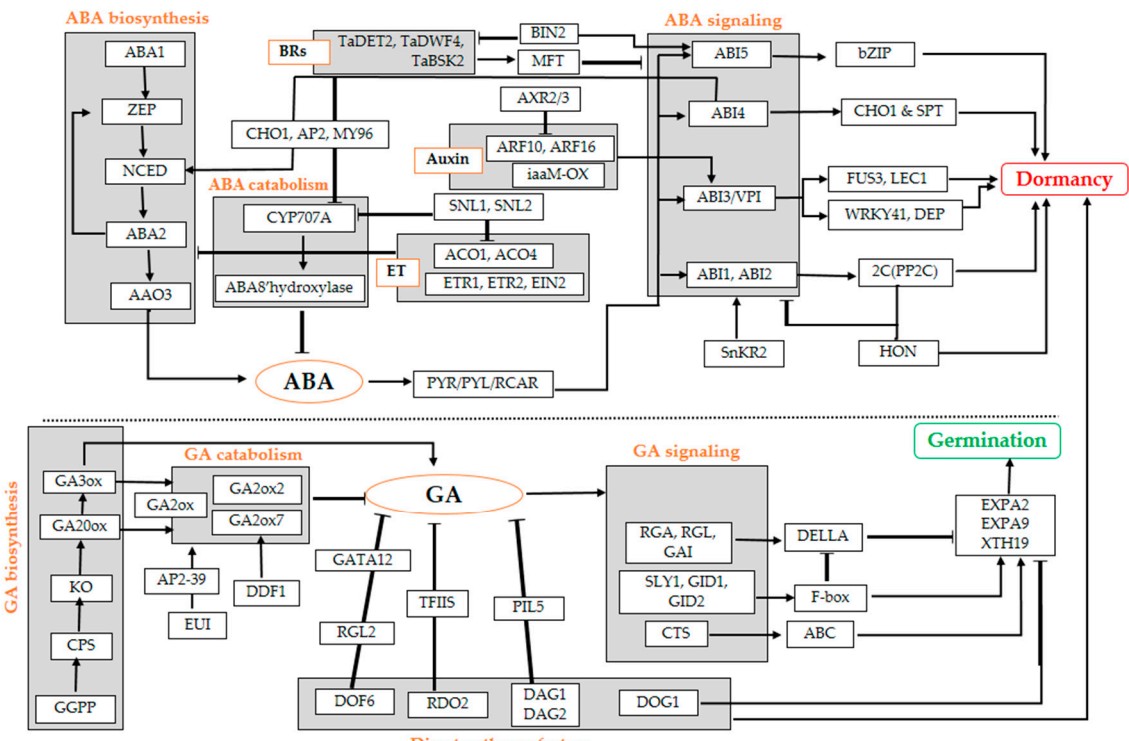

**Figure 2.** Mechanisms underlining seed dormancy and seed germination control in plants. This figure highlights abscisic and gibberellic acid metabolism and signalling pathways in plants and how ABA-GA balance is affected by the activity of other plant hormones and other genes enabling seed dormancy or seed germination. *ABA1*, abscisic acid deficient 1; *ABA2*, abscisic acid deficient 2; ZEP, zeaxanthin epoxidase; *NCED*, carotenoid cleavage dioxygenase; *AAO3*, abscisic aldehyde oxidase 3; ABA, abscisic acid; PYR/PYL/RCAR, pyrabactin resistance/pyrabactin-like/regulatory components of ABA receptors; *ABI*, abscisic acid insensitive; *VP1*, viviparous 1; bZIP, basic leucine zipper; *CHO1*, CHOTTO 1; *SPT*, SPATULA; *FUS3*, FUSCA3; *LEC1*, leafy cotyledon 1; WRKY41, WRKY DNA-binding protein 41; DEP, DESPIERTO; PP2C, protein phosphatase 2C; *HON*, HONSU; SnRK2, SNF1-related protein kinase 2; *CYP707A*, Cytochrome P450; ABA8′OH, ABA 8-hydroxylase; *AP2*, AP2-domain; *MYB96*, myeloblastosis 96; BRs, brassinosteroids; *TaDET2*, *Triticum aestivum* de-etiolated 2; *Ta DWF4*, *Triticum aestivum* DWF 4; *TaBSK2*, *Triticum aestivum* brassinosteroid signaling kinase 2; *BIN2*, brassinosteroid insensitive 2; *MFT*, mother of FT and TFL1; *ARF*, auxin-responsive factors; *AXR2*, auxin resistant 2; *AXR3*, auxin resistant 3; ACO, 1-aminocyclopropane-1-carboxylic acid oxidase; ET, Ethylene; *EIN2*, ethylene insensitive 2; *ETR1*, ethylene triple response 1; *ETR2*, ethylene triple response 2; *SNL1*, SIN3-like 1; *SNL2*, SIN3-like 2; GGDP, geranylgeranyl pyrophosphate; *CPS*, ent-copalyl diphosphate synthetase; *KO1*, ENT-kaurene oxidase 1; *GA3ox1*, gibberellins 3-oxidase 1; *GA20ox3*, gibberellins 20-oxidase 3; *GA2ox2*, gibberellins 2-oxidase 2; *GA2ox7*, gibberellins 2-oxidase 7; *AP2-39*, AP2 domain-containing transcription factor OsAP2-39; *EUI*, elongated uppermost internode; *DDF1*, delayed flowering 1; GA, gibberellins; GATA, GATA proteins; ABC, ATP-binding cassette; *DOF6*, DNA binding1 zinc finger 6; *DAG1*, DOF affecting germination1; *DAG2*, DOF affecting germination2; *DDF1*, delayed flowering 1; DELLA, DELLA proteins; F-box, F-box proteins;

*DOG1*, Delay of Germination 1; *RGA*, repressor of gibberellin acid; *RGL1*, repressor of gibberellin acid like 1; *RGL2*, repressor of gibberellin acid like 2; *RDO2*, reduced dormancy 2; *CTS*, comatose; *PIL5*, phytochrome interacting factor 3-like 5; *TFIIS*, transcription factor S-II; *SLY1*, sleepy1; *GAI*, gibberellin acid insensitive; *GID1*, gibberellin acid insensitive dwarf1; *GID2*, gibberellin acid insensitive dwarf2; *EXPA2*, expansin-A2; *EXPA9*, expansin-A9; *XTH19*, xyloglucan endo-transglycosylase 19.

### 4.1. Direct Pathway Regulation

Delay of Germination (*DOG*) genes are important determinants of seed dormancy within populations. The seed dormancy QTL called Delay of Germination 1 (*DOG1*) was identified as responsible for a strong dormancy and acting in interaction with *DOG3* [46]. *DOG1* was shown to be specifically expressed during seed development with detectable levels in dry seeds [47]. Global transcript analysis in *Arabidopsis* using microarrays indicated that the expression level of *DOG1* decreased during after-ripening [46]. The expression of *DOG1* was also reduced in the *hub1* mutant characterized by reduced dormancy in agreement with a role of *DOG1* in regulating dormancy levels [48]. Bentsink et al. [49] identified eleven *DOG* QTLs but there is the absence of strong epistasis interaction between different *DOG* loci suggesting that *DOG* loci affect dormancy via distinct genetic pathways. However, not all *DOG* genomic regions identified by Bentsink et al. [49] contain genes that have previously been associated with seed dormancy. For instance, *DOG3* collocates with *LEC2*, *DOG4* with *TT7* (Transparent Testa 7), DOG5 with *ABI1*, *DOG19* with *GA2*, *DOG20* with *PIL5*, and *DOG22* with *RGL2*. *DOG1* is involved in the enhancement of dormancy by low temperatures during seed maturation and is a central factor of seed dormancy [29,50–53]. Homologs of Arabidopsis *DOG1* were characterized in other plants including *Brassica napus* L. [54,55], rice [54,56], *Lepidium sativum* L. [57], barley, and wheat [58]. Ashikawa et al. [58] reported that *DOG1*-like genes in wheat and barley were good candidate transgenes for reducing pre-harvest sprouting in wheat.

The DNA BINDING1 ZINC FINGER (DOF) proteins are a family of plant transcription factors in the plant kingdom and were identified as potential regulators of seed dormancy [59,60]. In *Arabidopsis* species, the genes encoding transcription regulators such as DOF affecting germination (*DAG*) have been identified and investigated for their efficiency in seed dormancy regulation [59,61]. Stamm et al. [62] reported 36 DOF proteins in *Arabidopsis*, many of which have been implicated in the regulation of germination. To support this, the DOF zinc finger proteins (DOF Affecting Germination 1 and 2) have been shown to possess opposing roles in the regulation of germination. For instance, *DAG1* inhibits germination by mediating *PIL5* activity and affecting gibberellin biosynthesis [63,64]. *DOF6* was shown to negatively regulate germination by affecting abscisic acid signalling in seeds [60]. Recently, Ravindran et al. [61] reported a novel crosstalk between DOF6 and *RGL2* that enables primary dormancy in *Arabidopsis* through GATA transcription factor regulation. This novel *RGL2*–DOF6 complex is required for activating *GATA12*, a gene encoding a GATA-type zinc finger transcription factor, as one of the downstream targets of *RGL2* expression in *Arabidopsis thaliana*, thus revealing a molecular mechanism to enforce primary seed dormancy by repressing GA signalling [59,61,62].

Other important genes are those related to reduced dormancy (*RDO*). A mutagenesis screen for seed dormancy in *Arabidopsis* yielded reduced dormancy (*RDO*) mutants, which appeared to be central for the dormancy mechanisms and an important target for seed dormancy research [65,66]. For example, *RDO2*, one of the genes identified from the screening, encoded *TFIIS* [67] and a mutation in *TFIIS* resulted in reduced seed dormancy [68]. Mutants *dog1* and *rdo4* presented a reduced seed longevity phenotype [55,69,70]. Then, the reduced dormancy mutants *rdo1* and *rdo2* are not affected in their response to ABA.

### 4.2. Hormonal Pathway Regulation

Abscisic acid (ABA) produced by embryo is fundamental for the promotion of seed dormancy unlike that produced by maternal tissues through ABA biosynthesis genes such as carotenoid cleavage dioxygenase (*NCED*), ABA-deficient (*ABA1*, *ABA2*), and abscisic aldehyde oxidase 3 (*AAO3*) [71–74]. In *Arabidopsis thaliana*, the induction of seed dormancy is due to *NCED5*, *NCED6*, and *NCED9* genes

while *NCED1* and *NCED2* genes are reported regulating seed dormancy in cereals crops [17,75]. ABA is catabolized through ABA8′hydroxylase encoded by Cytochrome P450 (*CYP707A*) genes including *CYP707A1*, *CYP707A2* and *CYP707A3* [17]. ABA is signalled in *Arabidopsis* and monocot plants through ABA insensitive genes, such as *ABI1*, *ABI2*, *ABI3/VP1*, *ABI4*, *ABI5*, and *ABI8*. *ABI3/VP1* genes seem to be the most important in seed dormancy induction whereas *ABI4* regulates abiotic stress responses and different aspects of plant development and plays an important role in seed dormancy maintenance by binding itself to the promoter of *CYP707A1* and *CYP707A2* genes and repressing their expression [76]. Therefore, *NCED2* and *NCED3* activation is enhanced through *CHO1* encoding with APETELA 2 (*AP2*) domain, the ABA-responsive R3R2-type MYB transcription factor: myeloblastosis 96 (MYB96) [17,77]. These ABA signalling genes regulate seed dormancy through several transcription factors including Pyrabactin resistance proteins/PYR-like proteins/regulatory components of ABA receptor (PYR/PYL/RCAR), phosphatase 2C (PP2C), HONSU (HON), SNF1-related protein kinase 2 (SnRK2), and abscisic acid-responsive elements—binding factor (AREB) and basic leucine zipper (bZIP) [17,77–79].

Kucera et al. [80] identified an early and late GA biosynthesis within the embryo encoded respectively by ent-copalyl diphosphate synthetase (*CPS1*) gene and *GA3ox2* gene in the cortex and endodermis of the root. Over-expression of GA biosynthesis genes such as gibberellins 3-oxidase 1 (*GA3ox1*), gibberellins 20-oxidase 3 (*GA20ox3*), and ENT-kaurene oxidase 1 (*KO1*) results in seed germination [17,81,82]. The regulation of GA signalling gene comatose (*CTS*) is a key component to have dormancy or germination. Its activation results in germination through a peroxisomal protein of the ATP-binding cassette (ABC) transporter class [17,83]. DELLA proteins act as repressors of GA signalling and integrate environmental cues into GA signalling through the expression of the repressor of GA (*RGA*), RGA-like 1 (*RGL1*), RGA-like 2 (*RGL2*), gibberellic acid insensitive (*GAI*), and GA INSENSITIVE DWARF (*GID1*, *GID2*) genes [17,84]. The F-box protein is a receptor that mediates GA responses for the degradation of DELLA-type transcription repressors [80] through the activation of sleepy1 (*SLY1*) gene [17,85]. Contrarily, Delay of Germination 1 (*DOG1*) regulates the expression of GA biosynthesis genes by the inhibition of genes encoding cell wall remodelling enzymes and by regulating the appropriate time of germination according to ambient temperature [86]. Another GA catabolism genes include elongated uppermost internode (*EUI*) and delayed flowering 1 (*DDF1*). *ABI4* recruits an additional seed-specific transcription factor to repress the transcription of GA biogenesis gene or can directly bind itself to the promoter of *GA2ox3* as *ABI5*, activating its expression [76,87].

It is reported that auxin may suppress seed germination under high salinity [88], delay seed germination, and inhibit pre-harvest sprouting through indole-3-acetic acid: IAA [89]. Auxin synthesis decreases during after-ripening treatment enabling seed dormancy break [90] and helps ABA in dormancy induction and protection [91] by repressing, for example, the embryonic axis elongation during seed germination [92].

Ethylene (ET) participates in the seed dormancy regulation through its receptors such as ethylene triple response (*ETR1*, *ETR2*) and ethylene insensitive 2 (*EIN2*). ET may repress ABA accumulation and promote seeds dormancy release so that the high ET content in seeds is associated with dormancy loss through SIN3-like 1 (*SNL1*) and *SNL2* genes at the epigenetic level [17,77,93–95]. At the same time, *SNL1* and *SNL2* promote seed dormancy through another pathway [96].

Brassinosteroid (BR) action consists in suppressing the inhibitory effect of ABA during the germination process in wild types of *Arabidopsis* through an MFT (Mother of Flowering locus T (FT) and Terminal flower 1 (TFL1))-mediated pathway [97]. The lower content of BR during germination, due to *BIN2* (Brassinosteroid Insensitive 2) gene activation, stabilizes ABI5 protein to mediate ABA signaling unless there is BR treatment to repress the *BIN2–ABI5* interaction [17,98]. BR biosynthesis genes (TaDE-etiolated 2 (*TaDET2*) and *TaDWARF* 4) and BR signalling promotion genes (TaBR signalling kinase 2 (*TaBSK2*)) have been identified in wheat [17,99].

Jasmonic acid (JA) can inhibit the germination process promoting the effect of ABA biosynthesis genes regulatory action on *MFT* gene by 12-oxo-phytodienoic acid (OPDA) in dry seeds whereas during

imbibition the transcription of those genes is repressed by JA [17,77,100–102]. Salicylic acid (SA) in a first way inhibits the expression of GA-induced a-amylase genes under normal growth conditions [77,103]. In the other way, it reduces oxidative damage under high salinity [77,104]. Cytokinin (CTKs) might effectively concentrate and direct cell division and elongation of the emerged root. In sorghum, cytokinin/ABA interaction controls germination by inducing ABI5 protein degradation [80,105]. Strigolactones (SLs) are involved in seed germination in *Arabidopsis*, in parasitic weeds, and in other species [77,106]. ABA is involved in SLs biosynthesis regulation in tomatoes [107]. Furthermore, some key components in the SL signalling pathway affect seed germination, including *SMAX1* (Suppressor of More Axillary Growth2 1) in *Arabidopsis* and its homolog *OsD53* in rice [77,108].

Numerous studies of the natural variation and various mutants have offered an opportunity to identify new seed dormancy regulators, their related genes or QTLs, and how they work. The description of seed dormancy regulators used in this review are presented in Tables 3 and 4.

**Table 3.** Description of mutant genes controlling seed dormancy reported by previous studies in *Arabidopsis thaliana*.

| Mutants | Description/Action | References |
|---|---|---|
| *nced6*/*nced9* and *nced5* | Promote germination | [75,109,110] |
| *aao3*, *aba1*, and *aba2* | Reduce dormancy | [73,111] |
| *cyp707a* | Enhance seed dormancy level | [17,77,112,113] |
| *abi1* | Reduce dormancy through chilling and dry storage, reduce ABA sensitivity for germination and no precocious germination | [80,114] |
| *abi3* | Leads to seed dormancy even in immature seeds | [115–118] |
| *cts* | Leads to the seed dormancy protection even after stratification and after-ripening | [80] |
| *yuc1*/*yuc6* (Auxin) | Reduce seed dormancy | [89] |
| *ein2* (Ethylene) | Leads to higher expression of *NCED3* | [17,119,120] |
| *etr1* | Induces lower activation of *CYP707A2* genes | |
| *snl1* and *snl2* | Reduce seed dormancy together with the increased Ethylene content | [96] |
| *hub1*(*rdo4*) | Characterized by a reduced dormancy | [48] |
| *tfiis* | Reduces seed dormancy | [68] |
| *dog1* and *rdo4* | Reduce seed longevity phenotype | [55,69,70] |
| *rdo1* and *rdo2* | Not affected in their response to ABA | |

Note: *yuc1*/*yuc6*, mutants of Yucca flavin monooxygenase genes; *hub1*, mutant of Homologous to UBIquitin; *nced*, mutant of *NCED*; *aao3*, *AAO3* mutant; *aba1*/*aba2*, *ABA1*/*ABA2* mutant; *CYP707a*, *CYP707A* mutant; *abi1*/*abi3*, *ABI1*/*ABI3* mutant; *cts*, *CTS* mutant; *ein2*, *EIN2* mutant; *snl1*/*snl2*, *SNL1*/*SNL2* mutant; *tfiis*, *TFIIS* mutant; *dog1*, *DOG1* mutant; *rdo1*/*rdo2*/*rdo4*, *RDO1*/*RDO2*/*RDO4* mutant; *etr1*, *ETR1* mutant.

**Table 4.** Processes and genes involved in seed dormancy regulation.

| Process | Genes | Description | Related Species | References |
|---|---|---|---|---|
| ABA biosynthesis | *NCED5, NCED6, NCED9* | Induction of seed dormancy | *Arabidopsis thaliana* | [17,75,109,121,122] |
| | *NCED1, NCED2* | Induction of seed dormancy | *Oryza sativa, Hordeum vulgare* | [17] |
| | *ABA1, ABA2* | Encode for zeaxanthine poxidase | *Arabidopsis thaliana; Zea mays; Nicotiana plumbaginifolia* | [80] |
| | *AAO3* | Encodes final step of ABA biosynthesis | *Arabidopsis thaliana* | [80,123] |
| ABA catabolism | *CYP707A1, CYP707A2, CYP707A3* | Encode for ABA8'hydroxylase; loss of dormancy | *Arabidopsis. thaliana, Hordeum vulgare* | [17] |
| ABA signalling | *ABI1, ABI2* | Encode for Serine/threonine phosphatase 2C (PP2C) inducing seed dormancy | *Arabidopsis thaliana* and monocot | [80,114] |
| | *ABI3/VP1* | Regulation of chlorophyll, anthocyanin, and storage proteins accumulation with *FUS3* and *LEC1* | *Arabidopsis thaliana* and monocot | [17,124,125] |
| | | Regulated by *WRKY41* and by *DEP* for primary seed dormancy establishment | *Arabidopsis thaliana* and monocot | [77,126,127] |
| | *ABI4* | Regulated by transcription factors CHO1 and SPT for dormancy establishment and maintenance through *NCED2* and *NCED3*; Represses *CYP707A1* and *CYP707A2* | *Arabidopsis thaliana* and monocot | [77,128–130] |
| | *ABI5* | Regulated by *bZIP* transcription factor for positive ABA signalling and repressing seed germination | *Arabidopsis thaliana, Sorghum bicolor* | [77,131] |
| GA biosynthesis | *GA3ox1, GA20ox3, KO1* | Inducing of hydrolytic enzymes that weaken the seed coat, inducing of mobilization of seed storage reserves, and stimulating of expansion of the embryo | *Arabidopsis thaliana* and monocot | [17,81,82] |
| | *CPS* | Catalyzed geranylgeranyl pyrophosphate (GGDP) cyclization reaction in the provascular tissue | *Arabidopsis thaliana* | [80] |
| GA signaling | *CTS* | Encodes a peroxisomal protein of the ATP-binding cassette (ABC) transporter class | *Arabidopsis thaliana* | [17,83] |
| | *RGA, RGL1, RGL2, GAI* | Encode DELLA proteins as a repressor of GA signalling | *Arabidopsis thaliana* | [17,79,80,84] |
| | *SLY1* | GA relieves DELLA repression of seed germination by F-box protein | *Arabidopsis thaliana* | [17,85] |
| | *GID1* | Induce release of seed dormancy by promoting interaction of DELLA with the F-box protein | *Arabidopsis thaliana, Oryza. sativa* | [81,132,133] |
| | *GID2* | Encodes for F-box subunits of an SCF E3 ubiquitin ligase that ubiquitinates DELLA proteins | *Arabidopsis thaliana, Oryza sativa* | [81,134–137] |
| GA catabolism (GA2ox2) | *DOG1* | Inhibition of genes encoding cell wall remodelling enzymes: *EXPA2, EXPA9, XTH19* by regulates the expression of GA biosynthesis genes | *Arabidopsis thaliana* and monocot | [86] |
| | DDF1 | Promotes transcription of the GA inactivation gene *GA2ox7* | *Arabidopsis thaliana* | [77,138] |
| | EUI | Promoted by AP2 domain-containing transcription factor *OsAP2-39* for GA inactivation | *Oryza sativa* | [77,139] |

**Table 4.** *Cont.*

| Process | Genes | Description | Related Species | References |
|---|---|---|---|---|
| Auxin | *iaaM-OX* | Strong seed dormancy | *Triticum aestivum* | [89] |
| | *ARF10* and *ARF16* | Activates *ABI3* by perceiving high level of IAA for dormancy maintenance | *Arabidopsis thaliana* | [17,77,91] |
| | *AXR2/3* | Repress *ARF10* and *ARF16* | *Arabidopsis thaliana* | |
| Ethylene | *ACO1, ACO4* | Ethylene biosynthesis genes | *Arabidopsis thaliana* | [17,140] |
| | *ETR1, ETR2 EIN2* | Contrasting roles for ABA biosynthesis during seed germination under salt-stress conditions | *Arabidopsis thaliana* | [141] |
| | *SNL1* and *SNL2* | Reduce acetylation level of histone 3 lysine 9/18 and histone 3 lysine 14 repressing ABA accumulation at high level of ET | *Arabidopsis thaliana* | [96] |
| | *SNL1* and *SNL2* | Promote seed dormancy through simultaneous modulation of *ACO1, ACO4* and *CYP707A1, CYP707A2* | *Arabidopsis thaliana* | [96] |
| Brassinosteroid biosynthesis | TaDE-etiolated 2 (*TaDET2*) and *TaDWARF 4* | Ensure BR production in plant | *Triticum aestivum* | [17,99] |
| Brassinosteroid signaling | TaBR signalling kinase 2 (*TaBSK2*) | Promote BR signalling | *Triticum aestivum* | [17,99] |
| | *MFT* | Forming a negative feedback loop to modulate ABA signalling | *Arabidopsis thaliana* | [142,143] |
| | *BIN2* | Key repressor of the BR signalling | *Arabidopsis thaliana* | [17,98] |
| Jasmonic acid | OPDA | Promote effect of *ABA1, ABI5*, and *RGL2* and its regulatory action on *MFT* gene for seed dormancy maintenance | *Arabidopsis thaliana* | [17,77,100,144]. |
| Other genes | *DOG1* | Shows strong dormancy | *Arabidopsis thaliana, Hordeum vulgare, Triticum aestivum* | [29,50–53,55] |
| | *DAG1* and *DAG2* | Inhibiting germination by mediating *PIL5* activity as well as directly affecting gibberellin biosynthesis | *Arabidopsis thaliana* | [63,64] |
| | *DOF6* | Negatively regulates germination by affecting abscisic acid signalling in seeds | *Arabidopsis thaliana* | [60] |
| | *RDO2* | Encodes *TFIIS* for strong dormancy | *Arabidopsis thaliana* | [67] |
| | *GATA12* | Encodes a GATA-type zinc finger transcription factor for novel *RGL2–DOF6* complex enforcing primary seed dormancy via GA signalling repression | *Arabidopsis thaliana* | [59,61,62] |
| | NR (Nitrate reductase) | Promotes dormancy release | *Arabidopsis thaliana* | [81,145] |

## 5. Environmental Factors Influencing Seed Dormancy Regulation

After-ripening, stratification, light, and seed components influence seed germinability. After-ripening refers to the transition period from a dormant to a more readily germinable seed, where the seed is submitted to a set of environmental conditions after maturation and separation from the mother plant [146]. For instance, in *Gynandropsis gynandra*, this period is supposed to last about three to six months after harvest [31].

Temperature affects ABA-GA balance resulting in either germination or strong dormancy [51,77,147]. In wheat, seed germination was highly affected by genotype and temperature. Seed dormancy increased when seeds were developed under 15 °C whereas seed germination increased when seeds were sown at 20 °C [148–150]. For some African traditional leafy vegetables, the maximum germination occurs at 29 to 36 °C with 36 °C for *Vigna unguiculata* L., 35 °C for *C. olitorius*, and 30 °C for *G. gynandra* [33,39].

Light also plays a crucial role in seed germination by inducing a secondary dormancy in imbibed after-ripened seeds, increasing the expression of GA biosynthesis genes and repressing the expression of GA catabolism gene through the action of phytochrome [151,152]. For instance, red (R) light inhibits the expression of *NCED6*, fared (FR) light inhibits the expression of *CYP707A2*, and blue light enhances the transcription of ABA biosynthetic genes *NCED1* and *NCED2* [77,153,154]. While germination of *G. gynandra* increase under dark conditions [39], light is supposed to positively affect germination of TLVs such as *Amaranthus cruentus* L., *Brassica rapa* L. subsp. *chinensis*, *Corchorus olitorius*, *Citrullus lanatus* Thunbs., and *Solanum retroflexum* Dun. [33]. It is therefore important not only to assess the temperature and light requirements during seed development, seed storage, and seed germination for TLVs seed germination but also to explore the induced changes at the hormonal level. Ochuodho and Modi [155] reported negative photosensitivity at 20 °C in continuous white light during the germination of *G. gynandra*, probably controlled by changes in ABA and GA content. In addition, genetic and physiological studies of seed dormancy highlighted the effect of interactions between light and temperature on seed dormancy regulation in *A. thaliana* [156]. Some of the relations between environmental factors and seed dormancy regulators in plants reviewed by Skubacz and Daszkowska-Golec [17] are summarized in Table 5.

**Table 5.** Description of Environment factors and their role in seed dormancy regulation in plants reviewed by Skubacz and Daszkowska-Golec [17].

| Environment Factors | Situations | Role in Seed Dormancy Regulation | Description | Species |
|---|---|---|---|---|
| After-ripening | Seed dry storage period at room temperature | Reduced dormancy | Positive relationship with CYP707A2 Induces GA insensitive dwarf1 *GID1b* | *Arabidopsis thaliana* |
| | | | Promotes expression of JA biosynthesis genes: Allene oxide synthase (*AOS*), 3-ketoacyl coenzyme A (*KAT3*) and Lipoxygenase 5 (*LOX5*); Induces *GA20ox1* and *GA3ox2* | *Triticum aestivum* |
| | | | Increases the expression of *ABA8′OH-1* | *Hordeum vulgare*, *Brachypodium distachyon* |
| Temperature | Low temperature | Reduced dormancy | Promotes *GA3ox1* expression; Represses *GA2ox2* gene | *Arabidopsis thaliana* |
| | | Higher level of dormancy during seed development | Activates *MFT* gene | *Triticum aestivum* |
| | High temperature | Increased dormancy during seed imbibition | Represses *GA20ox1*, *GA20ox2*, *GA20ox3*, *GA3ox1*, and *GA3ox2* genes; Promotes the expression of ABA biosynthesis genes | *Arabidopsis thaliana* |
| Light | Red (R) light | Reduced dormancy | Inhibits the expression of *NCED6* | *Arabidopsis thaliana* |
| | Fared (FR) light | Increase dormancy | Inhibits the expression of *CYP707A2* | |
| | Blue light | Increased dormancy | Promotes *NCED1*, *NCED2*, *GA2ox3* and *GA2ox5* genes; Represses *GA3ox2* | *Hordeum vulgare* |

## 6. Seed Coat Components

Seed coat components functions are related to flavonoids, which provide greater mechanical restraint and reduced permeability to water, gases, and hormones. Flavonoids can inhibit metabolic processes of after-ripening and germination. They provide protection from oxidative damage and the dehydration and desiccation tolerance of orthodox and recalcitrant seeds, which correlated with ABA and active oxygen species (AOS), respectively [81,157–159]. The seed coat or pod colour may be a good indication for seed physiological status and can provide insight on seed germination as reported in *G. gynandra* [41]. Adebo et al. [160] reported two colours for seed coat (black and brown) among *Corchorus olitorius* cultivars from Benin. In amaranth species, a *MYB*-like transcription factor gene controls the seed colour variation especially between the ancestor, which is black, and the domesticated species, which is white [161]. Most of the seed dormancy constraints observed in TLVs seeds are due to the seed coat structure. This may be probably due to the important content of flavonoids observed in TLVs species whether in leaf, shoot, and seed. Table 6 informs about flavonoid content in the seed and shoot of some African leafy vegetable species.

**Table 6.** Flavonoids content in some TLVs species. Values collected from Akubugwo et al. [162], Paśko et al. [163], Yang et al. [164].

| Species | Part | Flavonoid (mg/100 g) |
|---|---|---|
| *Gynandropsis gynandra* | Shoot | 64.3 |
| *Corchorus olitorius* | Shoot | 63.9 |
| *Solanum nigrum* | Seed | 1.01 |
| *Amaranthus cruentus* | Seed | 667 |
| *Chenopodium quinoa* Willd. | Seed | 2238 |

## 7. Pathway for Dormancy Studies in Traditional Leafy Vegetables

Genetic and molecular approaches help find out the mechanisms underlying each step in the life cycle of plants [165]. The quantitative nature of seed dormancy should help identify whether seed dormancy of TLVs is controlled by nuclear or maternal factors or both. It is also important to know if the seed dormancy is genetically dominant or recessive or an outcome of a mutation. Factors that regulate germinability of seeds and their relationship for seed development and germination need to be investigated at different levels. Apart from temperature and light requirement, ecological significance of seed dormancy mechanism for TLVs should be understood. The current knowledge of dormancy established in well-studied crops such as *A. thaliana* and cereals will serve as baseline information to provide guidance for dormancy studies in TLVs. Therefore, researchers should be able to retrieve and put forward appropriate methodologies including definition of objectives and hypotheses, type of plant material (shoot, seed, source), conditions around implementation (seed development, seed storage, seed pre-treatments, seed germination), data collection, and analysis to perform genetic, genomic, and physiological screening of seed dormancy in TLVs. Studies can be specifically directed toward identification of the main seed dormancy regulators such as genes (*DOG1*, *DAG*, *DOF*, *RDO*, *ABI3*, *GA3ox1*, *GA2ox3*), transcription factors and proteins (ABA8'hydroxylase, CHO1, DELLA, GATA), abscisic–gibberellin acids crosstalk, non-enzymatic processes, seed morphological and structural components (endosperm, pericarp, embryo, seed coat), and the external factors (light, temperature, salinity, acidity, soil nitrate).

Seed characterization for a given TLV species will help identify the dormant and non-dormant genotypes and develop genetic populations for further studies. Figure 3 illustrates the conceptual approach leading to complete successful seed dormancy study in TLVs. This approach includes six steps from germplasm collection to the development of new varieties with high-quality seeds.

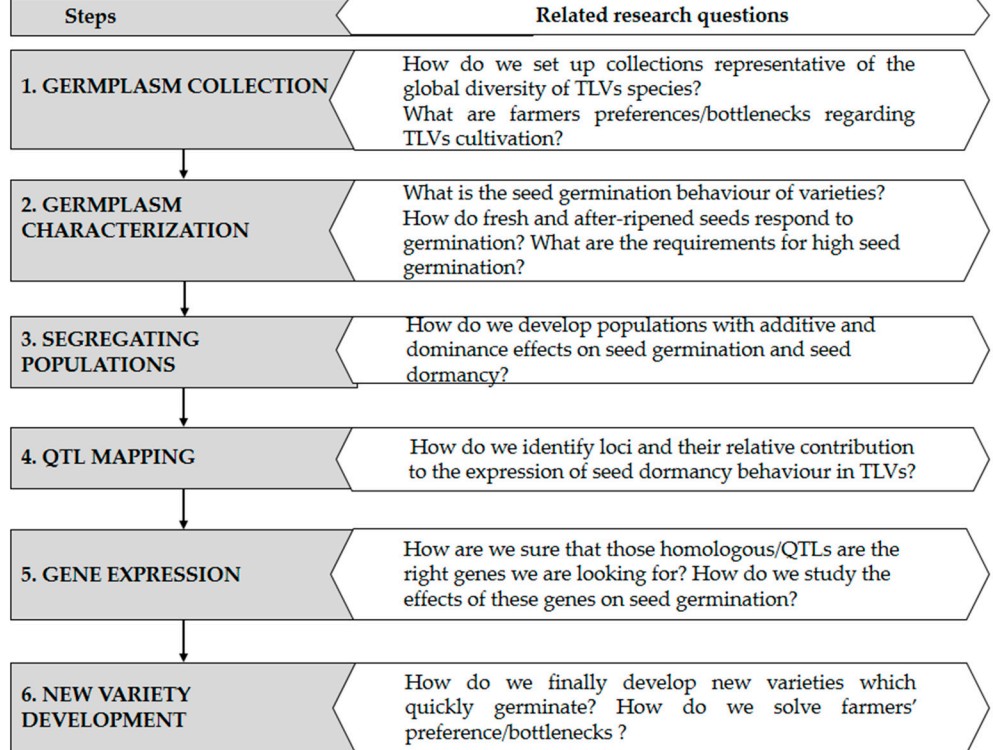

**Figure 3.** A conceptual approach for seed dormancy study in traditional leafy vegetables (TLVs) species. This proposed research avenue includes germplasm collection, germplasm characterization; development of mapping populations; quantitative trait loci (QTL) mapping; gene expression analysis; and new variety development. Germplasm collection is required in the range of distribution of the species for a developed core collection with cultivated and wild genotypes. Characterization of different factors influencing seed physiology and dormancy should help establish optimal germination conditions for fresh and after-ripened seeds and assess the natural variation of seed dormancy among genotypes. Development of mapping populations like biparental recombinant inbred lines (RILs), near-isogenic lines (NILs), F2, double haploid (HD), mutants, and multi-parental populations (multiparent advanced generation intercross (MAGIC)), multiparent recombinant inbred lines (AMPRIL)) will help access the genetic background existing among genotypes for seed dormancy. Development of molecular markers and use of modern genomic tools will lead to the successful implementation of mapping approaches such as in silico mapping, Bayesian mapping, linkage disequilibrium, multiple-QTL mapping, interval mapping, single marker-based, and marker regression for TLVs species. Tools such as expressed sequence tags (ESTs), serial analysis of gene expression (SAGE), massive parallel signature sequencing (MPSS), comparative genomics, microarray, and genome-wide association selection (GWAS) should be used for gene expression analysis. The development of improved varieties with high germination ability and many interest traits for farmers will be possible through Marker-assisted gene introgression, ultra-high-density bin map, and marker-assisted gene pyramiding.

*7.1. Germplasm Collection*

The use of wild and cultivated populations should enhance the possibility to identify new regulators with ecological relevance. The majority of TLV species are still neglected and underutilized and are poorly represented or even not present in gene banks. The first important task to achieve will therefore be to conduct germplasm collection missions to assemble as much as possible the natural diversity in these species by considering the geographical range of distribution of each given species. Seeds, roots, leaves, or any part of the reproductive material of plant should be collected in both wild and cultivated populations, and made available for scientists and gene bank curators. A good example is the *Gynandropsis gynandra* core collections developed by the World Vegetable Center, National

Plant Germplasm System of the United States Department of Agriculture, and the Laboratory of Genetics, Horticulture and Seed Science at University of Abomey-Calavi-Benin [11]. Such initiatives will contribute to the preservation of the genetic resources of TLV in the current context of increasing genetic erosion due to climate change, urbanization, and industrialization.

### 7.2. Seed Dormancy Characterization

Once germplasm collections are established or expanded, phenotypic characterization and genotypic characterization of seed dormancy related traits should be carried out. Phenotypic characterization includes variation in seed morphology and intrinsic germination capacity (radicle protrusion, seed, and seedling vigour). Seeds must be collected or developed at the same time for phenotypic characterization. Many studies of this type were conducted focusing on understanding the germination behaviour of various collections, comparisons of fresh dormant seed versus after-ripened non-dormant seeds, identifying light and temperature requirements, assessing the effects of various seed pre-treatments and seed origin [31,149,166]. These various experiments will lead to identifying populations with non-dormant seeds, populations with rather dormant seeds, and others with strong dormant seeds, in terms of germination rate, or time for dry storage to reach 50% germination or the number of days of seed dry storage required to reach 100% germination [49,167]. The genetic or molecular characterization of seed dormancy in mapping populations will be achieved using molecular markers as microsatellites or simple sequence repeats (SSR) and single nucleotide polymorphisms (SNPs). The combination of the data generated from the characterization will help predict the germination potential of the genotypes at an early stage of seed development. The observed variation about seed dormancy among populations could also be used to study gene expression profiles in different types of seeds. For instance, genetic and molecular dormancy investigations in *Arabidopsis* were performed using the reference accessions Landsberg erecta (Ler) and Colombia (Col) with low dormancy, the dormant accession Cape Verde Islands (Cvi), and various accessions with different levels of seed dormancy such as Antwerpen (An-1), St. Maria de Feira (Fei-0), Kondara (Kond), Shakdara (Sha), and Kashmir (Kas-2) [18,46,49,168]. Kępczynski et al. [36] have worked with primary dormant *Amaranthus retroflexus* seeds, secondary dormant *Amaranthus paniculatus* seeds, and non-dormant *Amaranthus caudatus* seeds.

### 7.3. Development of Mapping Populations for Identification of Candidate Genes Involved in Seed Dormancy

Seed dormancy is a quantitative trait with agronomic importance. Various segregating populations reviewed by Keurentjes et al. [169] have been used for QTLs identification, namely recombinant inbred lines (RILs), Near-Isogenic Lines (NILS), Multiparent Advanced Generation Intercross (MAGIC), and Multiparent Recombinant Inbred Line (AMPRIL). Thus far, RILs, NILs, or backcross inbred lines populations developed via crossing dormant and non-dormant accessions have been used in many seed dormancy studies [49,170]. Mutant lines were developed referring to transgenic techniques through overexpression or downregulation of genes and compared with wild types [54,58,171,172]. Doubled haploids, recombinant inbred lines, and near-isogenic lines are the most important types of population used in genetic mapping, gene discovery, and genomics-assisted breeding [173]. $F_2$ populations appear to be more powerful to help estimate both additive and dominance effects. Similar mapping populations should be developed for TLVs through mating systems such as diallel, nested, and factorial designs enabling to also assess the genetic variances and heritability of dormancy related trait.

The aim of developing mapping populations is to identify candidate loci and their relative contribution to the expression of seed dormancy behaviour. Several approaches have been developed for accuracy in QTLs detection such as single marker-based approaches, interval or LOD mapping, composite interval mapping, multiple interval mapping/multiple regression, marker regression, multiple-QTL mapping (MQM), Bayesian mapping, linkage disequilibrium mapping, meta-analysis, and in silico mapping [173,174]. For instance, the single marker regression and multiple-QTL mapping have been used in seed dormancy data analysis for *Arabidopsis thaliana* [18]. The heritability of major

QTLs through offspring will be important in the achievement of the breeding programs developed for traditional leafy vegetables.

The identified QTLs should help identify genes involved in cell organization and biogenesis, proteolysis, ribosomal proteins, hormones, elongation, and initiation factors that control seed dormancy in TLVs. Post-genomic analyses in previous studies about dormancy yielded interesting results through the use of proteomics, transcriptomics, metabolomics, and microarray analysis [175,176]. Their use should be focused on the comparison of transcription, post-transcription, gene expression either in imbibed seeds or fresh dormant seeds versus the non-dormant, after-ripened seeds. For example, a proteomic analysis of seed dormancy conducted in *Arabidopsis* with freshly harvested dormant seed and after-ripened non-dormant seed of accession Cvi revealed 71 proteins activities [177]; whereas 40 genes were reported by Li et al. [178] through the transcriptional profiling of imbibed *Brassica napus* L. seeds using *Arabidopsis* microarrays analysis. Various metabolic switches act during seed desiccation, vernalization, and early germination in *Arabidopsis* species [179] as well as in wheat [180]. In addition, genome-wide association analyses could provide a comprehensive understanding of gene activities in each of the three main regions of seed (coat, endosperm, embryo) during seed maturation, seed storage, and seed germination [181]. Comparative genomics will help construct strong phylogenies, identify changes in genome structure, annotate homologous genes, and understand novel traits [173,182]. Ayenan et al. [183] suggested the use of comparative genomics with rice to solve some of the constraints limiting the production of neglected African cereal crop fonio (*Digitaria* spp.). Comparative genomics is also relevant for TLVs genome screening as highlighted by Sogbohossou et al. [11] suggesting the use of available information on *Brassica* spp. and *Arabidopsis thaliana* genomes, *Solanum lycopersicum* and *Solanum tuberosum* L. genomes, and *Amaranthus hypochondriacus* L. and *Chenopodium quinoa,* respectively, for *Gynandropsis gynandra* and *Brassica carinata*, *Solanum* species (*Solanum nigrum*, *Solanum marcrocarpon*, *Solanum aethiopium* L., *Solanum scabrum*), and amaranths species (*Amaranthus* spp.).

*7.4. Tapping into Comparative Genomics to Study Seed Dormancy in TLVs*

*Arabidopsis thaliana*, as well as *Avena sativa, Solanum lycopersicum,* and *Nicotiana plumbaginifolia* have been used to understand many genetic and molecular determinants involved in germination and seed dormancy control for plants [175]. Recently, Sogbohossou et al. [11] presented spider plant as a good model plant for TLVs breeding studies. Spider plant is a promising vegetable species for food security and income generation for African communities in urban and in rural areas [184]. In addition, the genome size of spider plant is relatively small (~800 Mb) with the possibility to obtain three generations per year and offers many options in terms of breeding strategies [11]. Cleomaceae family and Brassicaceae family are closely related, sharing the At-β whole-genome duplication event within the Brassicales with fast molecular evolution rates [185]. Marshall et al. [186] argued the possibility to isolate orthologs of *Arabidopsis* genes from *Gynandropsis gynandra*, and van den Bergh et al. [187] reported that the genomes of *Gynandropsis gynandra* and *Arabidopsis thaliana* shared significant synteny, which will facilitate translational genomics between both species.

Seed dormancy is strongly reported in spider plant. Previous studies reviewed by Shilla et al. [31] support the assumption that spider plant accessions differ largely in their seed dormancy behaviour where the germination activation extends from zero (0) to more than six (6) months. Such a variation in the seed behaviour offers an opportunity to determine QTLs associated with germination in the species. *Gynandropsis gynandra* can be used as model plant for seed dormancy studies in plants the same way *Thellungiella salsuginea* has been used to investigate the physiological, metabolic, and molecular mechanisms of abiotic stress tolerance in plants [188]. Large physiological, genetic, and biochemical data should therefore be generated in spider plant for further investigations. As one of the priority species for the Mobility for Breeders in Africa project (in short "Mobreed") [189], some studies on *G. gynandra* are underway and will certainly provide us with more information about the potential of its genome. The use of comparative genomics, macrosynteny, and microsynteny is an alternative for the understanding of the seed dormancy determinants for *G. gynandra* and the generation of knowledge

regarding molecular markers, QTLs, candidate genes and their functions [190,191]. The development of DNA molecular markers should accelerate *G. gynandra* plant breeding as diversity arrays technology (DArT) and single nucleotide polymorphism (SNP) markers can be used to locate seed dormancy genes across *G. gynandra* germplasm [192]. Once candidate genes are identified, the development of populations with different genetic backgrounds such as NILs, F2, or a backcross population are important for gene introgression and validation. Simple sequence repeats (SSRs) markers have been successfully used in this way for various crops. The multi-environment experiment is a factor related directly to the efficiency of the identified genes. The way of mutant development can also be explored for new allele identification compared to the wild phenotype.

The knowledge established with the model plant (*G. gynandra*) can be transferred to other TLVs species through the development of a platform integrating genomics, transcriptomics, phenomics, and ontology analyses [193]. For instance, Mutwil et al. [194] established whole-genome coexpression networks for Arabidopsis, barley, rice, Medicago, poplar, wheat, and soybean that may considerably improve the transfer of knowledge generated in *Arabidopsis* to crop species. The isolation and functional analysis of homologs/QTLs should clearly help characterize all the genes and understand their interaction with each other. The understanding of others genetic phenomena of economical traits such as late-flowering date, leaf yield, and disease resistance, should help develop cultivars combining several good characteristics for farmers. The construction of ultra-high-density genetic maps is therefore essential to perform gene transfer through marker-assisted gene introgression and marker-assisted gene pyramiding [195].

## 8. Conclusions

This review sheds light onto a better understanding of the regulation of seed dormancy and the main factors known to be involved in the control of seed dormancy at hormonal, transcriptomic, epigenetic, protein, and environmental levels. The abscisic-gibberellin acids (ABA-GA) balance is a key component involved in the indirect control of seed dormancy that determines whether seeds may germinate or not. Auxin and salicylic acid (SA) promote seed dormancy; jasmonic acid (JA), brassinosteroids (BRs) and ethylene (ET) play a dual role in seed dormancy regulation, and cytokinin (CTKs) promotes seed germination. Other dormancy regulators were shown to directly promote or repress the decision of seeds to germinate. Those include genes such as Delay of Germination (*DOG*), *DOF* affecting germination (*DAG*), and reduced dormancy (*RDO*). For *Gynandropsis gynandra* and other TLVs, research is needed to solve the seed dormancy constraints and to provide prospective producers with high-quality seeds. *Gynandropsis gynandra*, a closely related species to *Arabidopsis thaliana*, was considered as a model plant to propose a pathway into solving the dormancy constraints in TLVs. This proposed research avenue includes germplasm collection, germplasm characterization, development of mapping populations, QTL mapping; gene expression analysis, and a new of variety development. Therefore, research is needed to highlight the specific storage conditions and seed pre-treatment required to ensure the seed viability and germination of genotypes. The mechanisms occurring during after-ripening in *G. gynandra* should also be elucidated. Rapid progress for *G. gynandra* full domestication should be achieved starting with the analysis of the natural variation in seed dormancy in the available genotype collections. Further steps include the characterization of the different factors influencing seed physiology, dormancy, and dormancy-breaking approaches. The identification of major genetic and molecular factors underlying seed dormancy during seed development, seed storage, and germination needs to be elucidated for developing cultivars that farmers can effectively use. Multidisciplinary research teams including physiologists, geneticists, and bioinformaticians are therefore required to quickly and efficiently make significant progress toward breeding for non-dormancy in traditional leafy vegetables.

**Author Contributions:** E.G.A.-D. and F.S.S. developed the conceptual framework of the manuscript. F.S.S. wrote the manuscript. H.P.F.Z., C.A.H., D.E.O.S., and E.G.A.-D. reviewed and approved the final version. All authors have read and agreed to the published version of the manuscript.

**Funding:** This work was supported by the Applied Research Fund of the Netherlands Organization for Science under the Project "Utilizing the genome of the vegetable species *Cleome gynandra* for the development of improved cultivars for the West and East African markets" (Project Number: W.08.270.350).

**Acknowledgments:** We thank the editors for the interactive improvement of the manuscript.

**Conflicts of Interest:** The authors declare no conflicts of interest.

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
