# Peer review of "Understanding Molecular Mechanisms of Seed Dormancy for Improved Germination in Traditional Leafy Vegetables: An Overview"

_agronomy, doi:10.3390/agronomy10010057_

Round 1

Reviewer 1 Report

The review on seed dormancy in African Traditional Leafy Vegetables by Sohindji et al., has the aim of providing some information on how to use genetics and environmental stimuli to solve the problem of uneven germination for many TLV due to an incorrect degree of seed dormancy.

The review deals with an agronomically interesting topic. I would suggest  to make more easily understandable the part on genetic and hormonal networks  so that it could be useful as a starting point for future projects for TLV breeding and to improve TLV- related agronomic practices.

Being TLV not studied at all, and also very linked to traditional agriculturl practices, I think that the review should be focused only on the major genes and pathways controlling it in model species, which would be the starting point for possible breeding programs. It should be as straight forward as possible, avoiding repetitions or minor details affecting dormancy in particular model species. The last paragraph (6) is quite useful to drive possible new researches.

While the first part of the manuscript (paragraphs 1-3) is quite focused and easy to follow, the second part (4-5) is often too detailed, confused and difficult to follow. I would suggest to reduce a bit these paragraphs, focus on major genes and factors and remove the repetitions. Paragraph 6 is also very long but sound.

DETAILED COMMENTS:

line 13- please explain already that TLVs are African traditional leafy vegetables, otherwise it is not clear until line 22

line 30. Please make an example,here or later for why TLVs are important to cope with climate change.

line 32-33. Please make some examples: which TLVs are only grown and "overexploited" in the wild; which are not yet fully domesticated? All of them?

line 53-54. Please revise the definition of dormancy, it is not vey clear. Dormancy is the capacity of a seed not to germinate under favourable conditions, and it can be controlled by genetic, hormonal, physiological and environmental factors...

line 65..."contrast" you mean, contrasting situations resulting from an inadeguate degree od sed dormancy?

lines 66-67: expressed: you mean present; over-expresed you mean high?

line 71: SEED dispersal

TABLE 1 is missing a legend. Also please be a bit more detailed, for example in the last row what do you mean for primary dormancy? too high? Please clarify all similar unclear cases in the table

LINE 125: moist storage - do you mean stratification (imbibition at low temperatures)? Stratification breaks dormancy in most  dormant species grown in temperate climates.

line 151: reserve accumulation occurs during seed development, please clarify; also desiccation tolerance needs to be acquired, also during seed germination.

line 161: abscisic ACID

FIGURE 1 is very complicated, including genes from different species...

Please always explain to which plant are you referring, genearally is Arabidopisis (which is NOT a crop), just remind it from time to time.

Line 219: DOF are not new, they regulate several developmental processes including, some of them, seed dormancy 

Line 238 (and other position in the paper) please write mutants in small letters (no capital) italic; genes capital italic, proteins capital.

LINE 274, if a gene is cited please add some info end references (this is an example)

LINE 281: repressing?

TABLE 2. very interesting table, column 1 should be called mutants instead of genes, and they should be italic; after the description of the mutant phenotype I would add the name of the species where they've been described.

TABLE 2: This is also very interesting:

in the title instead of genotipic factors I would write: processes and genes involved in dormancy. Instead of factors in the table I would write process. It would be helpful to add a column after genes with gene function.

Concerning the environmental factors I would also make a table, which helps a lot summarising also this part. Please list in this paragraph what is known concerning African TLVs.

LINE 336- seed coat components should not be in environmental factors I believe-Please make a separate paragraph and expand the information concening TLVs

line 344 cycle life...life cycle.

paragraph 6.2. It is vey important that seeds to be pheontyped for dormancy are collected at the same time in the varieties to be compared, as dormancy changes a lot during after ripening and depending on weather conditions.

Author Response

Answers to comments, revisions and questions of Reviewer 1

The review on seed dormancy in African Traditional Leafy Vegetables by Sohindji et al., has the aim of providing some information on how to use genetics and environmental stimuli to solve the problem of uneven germination for many TLV due to an incorrect degree of seed dormancy.

The review deals with an agronomically interesting topic. I would suggest to make more easily understandable the part on genetic and hormonal networks so that it could be useful as a starting point for future projects for TLV breeding and to improve TLV- related agronomic practices.

Being TLV not studied at all, and also very linked to traditional agricultural practices, I think that the review should be focused only on the major genes and pathways controlling it in model species, which would be the starting point for possible breeding programs. It should be as straight forward as possible, avoiding repetitions or minor details affecting dormancy in particular model species. The last paragraph (6) is quite useful to drive possible new researches.

While the first part of the manuscript (paragraphs 1-3) is quite focused and easy to follow, the second part (4-5) is often too detailed, confused and difficult to follow. I would suggest to reduce a bit these paragraphs, focus on major genes and factors and remove the repetitions. Paragraph 6 is also very long but sound.

Answer to the general comment: We deeply thank the reviewer for his positive comments on the manuscript and want to assure that all comments and suggestions will be carefully analysed and integrated in the revisions. Particularly we cut down the repetitive sections of the manuscript.

DETAILED COMMENTS:

Comment 1: line 13- please explain already that TLVs are African traditional leafy vegetables, otherwise it is not clear until line 22

Answer 1: We addressed the comment of the reviewer by precising that TLVs stand for African traditional leafy vegetables.  The first sentence of the abstract (lines 13-15) was edited as follows:

‘’Loss of seed viability, poor and delayed germination and inaccessibility to high-quality seeds are key bottlenecks limiting all year-round production of African Traditional Leafy Vegetables (TLVs).’’

We also defined traditional leafy vegetables (TLVs) at the first mention in the Introduction. 

Comment 2: line 30. Please make an example, here or later for why TLVs are important to cope with climate change

Answer 2: We added an example of drought tolerance of cleome and amaranth species. Amaranth species are widely used by local community as leafy vegetables. The first paragraph (line 32-39) of the introduction was edited as follows:

“Given the potential of African traditional leafy vegetables (TLVs) to cope with varying climate constraints and feed Africa, the production and consumption of the same have been promoted in the continent since the last two decade. For example, the antioxidant system of Cleome spinosa and Gynandropsis gynandra L. (Briq) cope with reactive oxygen species (ROS) formation under drought conditions limiting damage to cell structures, lipids, proteins, carbohydrates, nucleic acids, and cell death [5]. Luoh et al [6] reported also the lowest losses of nutrients in amaranth species and the African nightshade species when cultivating under deficient conditions.”

Uzilday, B., Turkan, I., Sekmen, A., Ozgur, R., Karakaya, H. Comparison of ROS formation and antioxidant enzymes in Cleome gynandra (C4) and Cleome spinosa (C3) under drought stress. Plant Sci. 2012, 182, 59-70.

Luoh, J.W., Begg, C.B., Symonds, R.C., Ledesma, D., Yang, R.-Y. Nutritional yield of African indigenous vegetables in water-deficient and water-sufficient conditions. Food Nutr Sci 2014, 5, 812.

Comment 3: line 32-33. Please make some examples: which TLVs are only grown and "overexploited" in the wild; which are not yet fully domesticated? All of them?

Answer 3: There are various management status of TLVs and sometimes their use might be related to sociolinguistic groups in such a way that the management status of a TLV can be wild in some sociolinguistic group while in others it will be cultivated. However, we added some examples of widely domesticated and semi-domesticated species. Please refer to the lines 39-43.

‘’Few species such as Amaranthus spp., Solanum scabrum Mill., and Solanum macrocarpon L. have been well domesticated, whereas Bidens pilosa L., Brassica carinata A. Braun., Gynandropsis gynandra L. (Briq), Corchorus spp., Launaea taraxacifolia Willd., Talinum triangulare Willd., etc. are widely used over the continent, but still semi-domesticated species and can grow spontaneously and/or cultivated according to the sociolinguistic groups or regions [3].’’

Comment 4: line 53-54. Please revise the definition of dormancy, it is not very clear. Dormancy is the capacity of a seed not to germinate under favourable conditions, and it can be controlled by genetic, hormonal, physiological and environmental factors…

Answer 4: Based on the reviewers’ comment, we edited lines 60-70 as follows:

“Seed dormancy is a state of a viable seed, expressed by the inhibition of germination under favourable environmental conditions required for adequate germination [12, 13]. It is an adaptive trait that optimizes the distribution of germination over time in a population of seeds [14]. On the other hand, germination is usually related to radicle protrusion, which is normally the visible result of germination [15]. Before this visible aspect, there are many events that begin with the uptake of water by mature dry seed, imbibition, and end with the embryonic axis elongation [16]. Seed dormancy, the non-viability of seed, and the absence of favourable environmental conditions for germination result in the absence of germination [12]. Non-germination due to unfavourable conditions is referred to as “quiescence” and enables seed survival for further seedling development under adverse conditions [17]. Seed dormancy appears as a complex quantitative trait under the influence of several genetic, hormonal, physiological and environmental factors [18].”

Comment 5: line 65..."contrast" you mean, contrasting situations resulting from an inadequate degree of seed dormancy?

Answer 5: We edited this section (lines 70-72) as follows:

“The pre-harvest sprouting and the absence or delay of germination after ripening are the two undesirable contrasting levels of seed dormancy.”

Comment 6: lines 66-67: expressed: you mean present; over-expressed you mean high?

Answer 6: Here “expressed” means “present” and “over-expressed” means “high level”. Lines 72-74 were edited as follows:

Consequently, the constraint about dormancy is twofold: either it is not present in seeds (zero level) leading to pre-harvest sprouting especially for cereals or it is present at high level leading to the absence or delay of germination at the desired time [19, 20].”

Comment 7: line 71: SEED dispersal

Answer 7: the word “Seed” have been added as follows (Line 78):

‘’…factors for germination after seed dispersal [22].’’

Comment 8: TABLE 1 is missing a legend. Also please be a bit more detailed, for example in the last row what do you mean for primary dormancy? too high? Please clarify all similar unclear cases in the table

Answer 8: The content of table 1 is written in clear and total letters with no abbreviation. We did not provide a legend for it. However, we clarified the sentence in the last row.

Comment 9: LINE 125: moist storage - do you mean stratification (imbibition at low temperatures)? Stratification breaks dormancy in most dormant species grown in temperate climates.

Answer 9: Moist storage treatment here stands for seeds imbibed in water and stored at low temperature (5°C). Yes, we meant ‘’stratification’’. So, the sentence was edited as follows (Lines 132-133):

“Dormancy breaking treatments with stratification…”

Comment 10: line 151: reserve accumulation occurs during seed development, please clarify; also desiccation tolerance needs to be acquired, also during seed germination.

Answer 10: We provided clarification about the link between reserve accumulation and seed development, desiccation tolerance and after ripening, and reserve mobilization and seed germination (Line 162-167)

“Seed dormancy is regulated by genotypic (internal regulation) and environmental factors during three stages in the persistent soil seed bank such as seed development, after-ripening, and seed germination. [46]. During seed development, some reserves are accumulated in seeds (reserve accumulation). During after-ripening, seeds especially orthodox seeds have ability to survive desiccation (desiccation tolerance). The seed germination stands for mobilization of reserves under favourable condition (reserve mobilization).”

Comment 11: line 161: abscisic ACID

Answer 11: This part has been deleted according to the comment

Comment 12: FIGURE 1 is very complicated, including genes from different species...

Please always explain to which plant are you referring, generally is Arabidopisis (which is NOT a crop), just remind it from time to time.

Answer to comment 11: We agree with the reviewer that Figure 1 may appear complicated to readers. That is the reason why we associated Table 3 to the figure. Each component inside the figure is detailed in Table 3 with appropriate species. Also more details were provided in the part 4 of the manuscripts. With Table 3 and Figure 1, readers can easily understand the information and knowledge provided.

Comments 13: Line 219: DOF are not new, they regulate several developmental processes including, some of them, seed dormancy

Answer 13:  The concerned part has been revised accordingly (Line 224)

“The DNA BINDING1 ZINC FINGER (DOF) proteins are a family of plant transcription…”

Comment 14: Line 238 (and other position in the paper) please write mutants in small letters (no capital) italic; genes capital italic, proteins capital.

Answer 14: The manuscript has been revised accordingly.

Comment 15: LINE 274, if a gene is cited please add some info end references (this is an example) 

Answer 15: The manuscript has been revised accordingly.

Comment 16: LINE 281: repressing?

Answer 16: the sentence has been corrected as follows (line 283-286):

“Auxin synthesis decreases during after-ripening treatment enabling seed dormancy break [91] and helps ABA in dormancy induction and protection [92] by repressing for example, the embryonic axis elongation during seed germination [93].”

Comment 17: TABLE 2. very interesting table, column 1 should be called mutants instead of genes, and they should be italic; after the description of the mutant phenotype I would add the name of the species where they've been described.

Answer 17: Table 2 has been improved accordingly. As the mutants were assessed in Arabidopsis species, we highlighted the name of species in the table title instead of adding special column for species (Line 315).

Table 2. Description of mutant genes controlling seed dormancy reported by previous studies in Arabidopsis thaliana

Comment 18: TABLE 2: This is also very interesting:

in the title instead of genotipic factors I would write: processes and genes involved in dormancy. Instead of factors in the table I would write process. It would be helpful to add a column after genes with gene function.

Answer 18: It is Table 3 not Table 2. We have corrected this. Also the description column already provides information related to gene function.

Comment 19: Concerning the environmental factors I would also make a table, which helps a lot summarising also this part. Please list in this paragraph what is known concerning African TLVs.

Answer 19: The section “5. Environmental factors influencing seed dormancy regulation” have been improved according to the reviewer comments and a table was added as suggested. The section was edited as follows:

5. Environmental factors influencing seed dormancy regulation

After-ripening, stratification, light and seed components influence seed germinability. After-ripening refers to the transition period from a dormant to a more readily germinable seed, where the seed is submitted to a set of environmental conditions after maturation and separation from the mother plant [150]. For instance, in Gynandropsis gynandra this period is supposed to last about three to six months after harvest [31].

Temperature affects ABA-GA balance resulting in either germination or strong dormancy [52, 78, 151]. In wheat, seed germination is highly affected by genotype and temperature. Seed dormancy increased when seeds were developed under 15°C whereas seed germination increased when seeds were sown at 20°C [152-154]. For some African traditional leafy vegetables, the maximum germination occurs at 29 to 36°C with 36°C for Vigna unguiculata L., 35°C for C. olitorius, and 30°C for G. gynandra [34, 40].

Light also plays a crucial role in seed germination by inducing a secondary dormancy in imbibed after-ripened seeds, increasing the expression of GA biosynthesis genes and repressing the expression of GA catabolism gene through the action of phytochrome [155, 156]. For instance, red (R) light inhibits the expression of NCED6, fared (FR) light inhibits the expression of CYP707A2, and blue light enhances the transcription of ABA biosynthetic genes NCED1, NCED2 [78, 157, 158]. While germination of G. gynandra increase under dark conditions [40], light is supposed to positively affect germination of TLVs such as Amaranthus cruentus L., Brassica rapa L. subsp. Chinensis, Corchorus olitorius, Citrullus lanatus Thunbs., and Solanum retroflexum Dun. [34]. It is therefore important not only to assess the temperature and light requirements during seed development, seed storage, and seed germination for TLVs seed germination but also to explore the induced changes at the hormonal level. Ochuodho and Modi [159] reported a negative photosensitivity at 20°C in continuous white light during the germination of G. gynandra, like controlled by changes in ABA and GA content. In addition, genetic and physiological studies of seed dormancy highlighted the effect of interactions between light and temperature on  seed dormancy regulation in A. thaliana [160].

The table 4 summarizes some of relation between environment factors and seed dormancy regulators in plants reviewed by Skubacz and Daszkowska‐Golec [17].

Table 4: Description of Environment factors and their role in seed dormancy regulation in plants reviewed by Skubacz and Daszkowska‐Golec [17]

Environment factors

Situations

Role in seed dormancy regulation

Description

Species

After-ripening

Seed dry storage period at room temperature

Reduced dormancy

Positive relationship with CYP707A2

Induces GA insensitive dwarf1 GID1b

Arabidopsis thaliana

Promotes expression of JA biosynthesis genes: Allene oxide synthase (AOS), 3‐ketoacyl coenzyme A (KAT3) and Lipoxygenase 5 (LOX5);

Induces GA20ox1 and GA3ox2

Triticum aestivum

Increases the expression of ABA8’OH-1

Hordeum vulgare, Brachypodium distachyon

Temperature

Low temperature

Reduced dormancy

Promotes of GA3ox1 expression;

Represses GA2ox2 gene

Arabidopsis thaliana

Higher level of dormancy during seed development

Activates MFT gene

Triticum aestivum

High temperature

Increased dormancy during seed imbibition

Represses GA20ox1, GA20ox2, GA20ox3, GA3ox1, and GA3ox2  genes;

Promotes the expression of ABA biosynthesis genes

Arabidopsis thaliana

Light

Red (R) light

Reduced dormancy

Inhibits the expression of NCED6

Arabidopsis thaliana

Fared (FR) light

Increase dormancy

Inhibits the expression of CYP707A2

Blue light

Increased dormancy

Promotes NCED1, NCED2, GA2ox3 and GA2ox5  genes;

Represses GA3ox2

Hordeum vulgare

Comment 20: LINE 336- seed coat components should not be in environmental factors I believe-Please make a separate paragraph and expand the information concerning TLVs

Answer 20: We thank the reviewer for this recommendation. It has been taken into account in the revised manuscript. The Seed coat components part is edited as follows:

Seed coat components

Seed coat components functions are related to flavonoids which provide greater mechanical restraint and reduced permeability to water, gases, and hormones. Flavonoids can inhibit metabolic processes of after-ripening and germination, and provide protection from oxidative damage, and the dehydration and desiccation tolerance of orthodox and recalcitrant seeds which correlated with ABA and active oxygen species (AOS) respectively [83, 161-163]. The seed coat or pod colour may be a good indication for seed physiological status and can provide insight on seed germination as reported in G. gynandra [40]. Adebo et al [164] reported two colours for seed coat (black and brown) among Corchorus olitorius cultivars from Benin. In amaranth species, a MYB-like transcription factor gene is controlling the seed colour variation especially between the ancestor which is black and the domesticated species which is white [165]. Most of the seed dormancy constraints observed in TLVs seeds are due to the seed coat structure. This may be probably due to the important content of flavonoids observed in TLVs species whether in leaf, shoot and seed. Table 5 informs about flavonoid content in the seeds of some African leafy vegetables species.

Table 5: Flavonoids content in some TLVs species. Values collected from Akubugwo et al [166], Paśko et al [167], Yang et al [168]

Species

Part

Flavonoid (mg/100 g)

Gynandropsis gynandra

Shoot

64.3

Corchorus olitoruis

Shoot

63.9

Solanum nigrum

Seed

1,01

Amaranthus cruentus

seed

667

Chenopodium quinoa

seed

2238

Akubugwo, I., Obasi, A., Ginika, S. Nutritional potential of the leaves and seeds of black nightshade-Solanum nigrum L. Var virginicum from Afikpo-Nigeria. Pak J of Nutr 2007, 6, 323-326. Paśko, P., Sajewicz, M., Gorinstein, S., Zachwieja, Z. Analysis of selected phenolic acids and flavonoids in Amaranthus cruentus and Chenopodium quinoa seeds and sprouts by HPLC. Acta Chromatogr 2008, 20, 661-672. Yang, R.-Y., Lin, S., Kuo, G. Content and distribution of flavonoids among 91 edible plant species. Asia Pac. J. Clin. Nutr. 2008, 17, 275-279.

Comment 21: line 344 cycle life...life cycle.

Answer 21: the line has been corrected to normal (Line 391)

Genetic and molecular approaches help find out the mechanisms underlying each step in the life cycle of plants [166]

Comment 22: paragraph 6.2. It is vey important that seeds to be pheontyped for dormancy are collected at the same time in the varieties to be compared, as dormancy changes a lot during after ripening and depending on weather conditions.

Answer 22: We thank the reviewer for this recommendation and assure that the precision has been added in line 435-436

“Seeds must be collected or developed at the same time for phenotypic characterization.”

Reviewer 2 Report

Title: The title is succinct but miss represents the manuscript. The title is not really what the paper is about because pages 5-13 are a general review of the literature.

Introduction: This can be shortened considerably. Eliminate lines 30-35 and the paper should start – “More than 1000 species of wild plants are used by people in ….”.  Line 53-92 can be eliminated and the introduction ended with lines 93-100.

Eliminate Methods it is a review not an experiment

Section 3. This is the most important section with Table 1. The authors mention there are about 1000 species of “Traditional Leafy Vegetables”  (TLV) or wild plant species exploited for food and culinary variety. However in Table 1 they present (5) five species. I would like to point out that under Amaranths the reference is for Crassocephalum which is an Aster (Asteraceae) and not an amaranth. I would also like to see Table one expanded to include the recommendations for overcoming the seed dormancy in these particular species as reported in the cited references (except for Amaranth which the cited paper does not address). Table 1 should be followed by a species by species discussion of the nature of seed dormancy in these particular species as reported by each of the authors cited. What I would like to point out here that the citation are very limited. Usually a review paper reviews a large body of literature. This is not the case here. There are only 5 species discussed and 6 relevant publications (excluding 29 and 34) to these particular species. Pages 5-13 is an unnecessary review in this context and I do not really see the relevance to the TLVs. If seed dormancy is well characterized in the five species this table should be followed with review of the type of dormancy and relevant literature to that particular type of seed dormancy, not a review of all types of seed dormancy.

Sections 6-7 should follow Section 3. In section 6 the conclusion is:  very little is known about these wild type varieties including characteristics of their seed dormancy. The section then goes on to suggest an investigative framework or general protocol. The paper seems to be more of a preliminary organization to the development of an active research program than a summary of a large body of information. Even section 6 seems excessive. Understanding all of the nuances of the molecular biology of seed dormancy would be essential if the goal was to genetically engineer a wild type species that lacks seed dormancy. It seems to me the shortest and most practical way to improving germination in these wild types is collect a variety of seed from individuals and developing a low cost breeding program.

Conclusion The conclusions, needs to be more work. Conclusions should

a) list practical steps to overcoming seed dormancy using a breeding program

b) the take home message of understanding the different types of seed dormancy in these specific species.

Recommendations

The paper is an eclectic assortment of ideas and suggestions about seed dormancy which may or may not be related to the traditional leafy vegetables. Style, organization and the accuracy of the Table 1 has some problems (e.g. like citation that are not relevant). A shorter more terse version of the paper might be more helpful, i.e., A short introduction that states the issue (as stated in the current title), Table 1 expanded, a discussion of what is known about the mechanism specific related to seed dormancy in the species in Table 1. A shortened section on suggestions on how to better understand seed dormancy or practical ways in which to run a breeding program to get the desired characteristics. The bibliography is extensive and impressive but would need to be limited to the specific topic, i.e. TLVs. I do not recommend publishing this paper in its current form. I think the paper needs to be reconceived to address seed dormancy in the target species, not in the generalized world of seed dormancy. A shortened version of this paper as text in a paper reporting experimental results of a breeding program on the target species would be the most appropriate use. This paper either needs to be a generalized critical review of seed dormancy in plants that have a potential for domestication, or specifically address seed dormancy in the plants listed in Table 1, but it can’t be both.

Author Response

Answers to comments, revisions and questions of Reviewer 2

Comments 27: The paper is an eclectic assortment of ideas and suggestions about seed dormancy which may or may not be related to the traditional leafy vegetables. Style, organization and the accuracy of the Table 1 has some problems (e.g. like citation that are not relevant). A shorter more terse version of the paper might be more helpful, i.e., A short introduction that states the issue (as stated in the current title), Table 1 expanded, a discussion of what is known about the mechanism specific related to seed dormancy in the species in Table 1. A shortened section on suggestions on how to better understand seed dormancy or practical ways in which to run a breeding program to get the desired characteristics. The bibliography is extensive and impressive but would need to be limited to the specific topic, i.e. TLVs. I do not recommend publishing this paper in its current form. I think the paper needs to be reconceived to address seed dormancy in the target species, not in the generalized world of seed dormancy. A shortened version of this paper as text in a paper reporting experimental results of a breeding program on the target species would be the most appropriate use. This paper either needs to be a generalized critical review of seed dormancy in plants that have a potential for domestication, or specifically address seed dormancy in the plants listed in Table 1, but it can’t be both.

Answer 27: We are thankful to the reviewer for this relevant comments and useful to improve the manuscript. The manuscript took into account the opinion of the reviewer that the paper should include ideas and suggestions about seed dormancy in plants in general and specifically in traditional leafy vegetables. This study aimed to provide a synthesis of the current state of knowledge about seed dormancy in TLVs and seed dormancy control in well studied plants and to discuss how to transfer such knowledge into traditional leafy vegetables research in tropical areas. We basically based our analysis on the wide cultivated species (Spiderplant, African nightshade, jute mallow, amaranth species). As we are focusing on genotypic factors of seed dormancy, it will be fortunate to learn from those target species. However, they are still neglected and underutilized crops with very limited studies conducted in this area. In the same time, seed dormancy is limiting local production and consumption. We believe that based on available information about other plants, we will be able to find a way to face this constraint. That is the reason why this paper presents available information about seed dormancy in plants in general before devising a strategy for using such information in TLVs for which there is not much known. More importantly, the knowledge needs to be synthesized and disseminated to propels research about seed dormancy in TLVs.

Comment 28: The title is succinct but miss represents the manuscript. The title is not really what the paper is about because pages 5-13 are a general review of the literature.

Answer 28:  The title has been improved and now reads:

‘’Understanding molecular mechanisms of seed dormancy for improved germination in traditional leafy vegetables: an overview’’

Comment 29: This can be shortened considerablyEliminate lines 30-35 and the paper should start – “More than 1000 species of wild plants are used by people in ….”.  Line 53-92 can be eliminated and the introduction ended with lines 93-100.

Answer 29: this recommendation has been considered in the revised paper. The paper starts as follows:

More than 1000 species were recorded to be used by African rural communities for dietary diversity, medicine purpose, food traditions and cultural identity [1-4]. Given the potential of African traditional leafy vegetables (TLVs) to cope with varying climate constraints and feed Africa, the production and consumption of the same have been promoted in the continent since the last two decade.

Comment 30. Eliminate Methods it is a review not an experiment

Answer 30: A ‘’methods’’ section is important in research area and we believe that doing a review is also a research. Experiment and review work overlap in many steps including (1) ask a question, (2) do some research, (3) Formulated a hypothesis, (4) record and analyze observations or results, (5) draw a conclusion. Our expectation is to provide readers with information that help understand how the review has been done from the development of the conceptual framework to reviewed and approved final version. This will insure a robust repeatability of our search and analysis. We believe that scholars should also learn from a systematic way to carry out literature search and review in a scientific way. That is why we decided to publicly share our methods.

Comment 31. This is the most important section with Table 1. The authors mention there are about 1000 species of “Traditional Leafy Vegetables” (TLV) or wild plant species exploited for food and culinary variety. However, in Table 1 they present (5) five species. I would like to point out that under Amaranths the reference is for Crassocephalum which is an Aster (Asteraceae) and not an amaranth. I would also like to see Table one expanded to include the recommendations for overcoming the seed dormancy in these particular species as reported in the cited references (except for Amaranth which the cited paper does not address). Table 1 should be followed by a species by species discussion of the nature of seed dormancy in these particular species as reported by each of the authors cited. What I would like to point out here that the citation are very limited. Usually a review paper reviews a large body of literature. This is not the case here. There are only 5 species discussed and 6 relevant publications (excluding 29 and 34) to these particular species. Pages 5-13 is an unnecessary review in this context and I do not really see the relevance to the TLVsIf seed dormancy is well characterized in the five species this table should be followed with review of the type of dormancy and relevant literature to that particular type of seed dormancy, not a review of all types of seed dormancy.

Answer 31:

We thank the reviewer for this suggestion. In fact, the review was based on those five species not because they are well documented but because they are widely cultivated over the continent. Table 1 is supposed to show that seed dormancy is reported in those main cultivated TLVs where adoption is promoted. So far, references for amaranths has been added and the following paragraph (Line 128-150) provides some details related to each species to make more understandable the dormancy situation in each species. All this information could not be squeezed within a table. But we will appreciate the decision of the editor.

“Seeds of Waterleaf (Talinum triangulare) are known to exhibit a kind of dormancy due to the impermeability of the seed testa and some undetermined physiological factors [35, 36]. These authors reported that scarification, alternating temperatures (6-l0°C and 28-35°C) and constant temperature (34°C) should enhance germination of waterleaf seeds. Taab and Andersson [35] reported a deeper level of primary dormancy for nightshade (Solanum nigrum). Dormancy breaking treatments with stratification, potassium nitrate, and gibberellic acid failed to show encouraging results, and are often not applicable at farmers’ level [35, 38]. A loss of viability and a poor germination of fresh and old seeds in jute mallow (Corchorus olitorius) are associated with the impermeability of jute mallow seed coat [33, 34]. In the case of spider plant (Gynandropsis gynandra), its cultivation is limited by the fact that its seeds can exhibit a high dormancy lasting for several months. {Geneve, 1998 #198}Geneve [39] reported that a primary non-deep physiological dormancy occurs in spider plant while Ochuodho and Modi [40] suspected physical dormancy and secondary dormancy. Ekpong [41] clarified that spider plant seeds are permeable to water but this water is trapped in the tissue between the embryo and the seed coat creating an oxygen barrier. Baskin and Baskin [32] concluded that spider plant exhibits a physiological dormancy. Recently, Shilla et al [31] reported that there were no dormancy cases on fresh seeds of spider plant according to World Vegetable Center preliminary results. Nevertheless, the seeds can stay in dormant state for several months before germination is activated and improved with dry storage periods [42-44]. Various levels of seed dormancy such as primary and secondary dormancy occur among Amaranths species. For instance there are primary dormant Amaranthus retroflexus L., secondary dormant Amaranthus paniculatus L. and non-dormant Amaranthus caudatus L. seeds [37]. Seed treatments such as seed holding in low temperature, pre-chilling and the application of ethylene induce dormancy breaking and accelerate the germination process in Amaranthus seeds [45].”

Kępczynski, J., Bihun, M., Kępczynska, E., Ethylene Involvement in the Dormancy of Amaranthus Seeds, in Biology and Biotechnology of the Plant Hormone Ethylene, A.K. Kanellis, C. Chang, H. Kende, and D. Grierson, Springer Netherlands: Dordrecht.1997, pp. 113-122.

Enayati, V., Esfandiari, E., Pourmohammad, A., Haj Mohammadnia Ghalibaf, K. Evaluation of different methods in seed dormancy breaking and germination of Redroot Pigweed (Amaranthus retroflexus). Iran J Seed Res 2019, 5, 129-137.

Comment 32: should follow Section 3. In section 6 the conclusion is:  very little is known about these wild type varieties including characteristics of their seed dormancy. The section then goes on to suggest an investigative framework or general protocol. The paper seems to be more of a preliminary organization to the development of an active research program than a summary of a large body of information. Even section 6 seems excessive. Understanding all of the nuances of the molecular biology of seed dormancy would be essential if the goal was to genetically engineer a wild type species that lacks seed dormancy. It seems to me the shortest and most practical way to improving germination in these wild types is collect a variety of seed from individuals and developing a low cost breeding program.

Answer 32: Our goal is not to develop a genetically modified wild type as the reviewer suggest but rather to breed for non-dormant cultivars. We believe that developing informative molecular markers for dormancy screening will be easier if we have good knowledge of the genetic factors affecting dormancy in our target species, especially for the species for which a reference genome is available. We believe that the development of new cultivars is important and should address seed dormancy as well.

Comment 33: Conclusion The conclusions, needs to be more work. Conclusions should

a) list practical steps to overcoming seed dormancy using a breeding program b) the take home message of understanding the different types of seed dormancy in these specific species.

Answer to comment 33: We thank the reviewer for these suggestions. We edited the conclusion firstly by presenting the major identified seed dormancy regulators in plants. Afterwards, we alluded to Gynandropsis gynandra as model species illustrating possible research avenues with 6 mains steps for seed dormancy analysis in TLVs. In addition, we listed some practical research points required for Gynandropsis gynandra species and ended by inviting researchers from multidisciplinary areas to generate relevant knowledge for non-dormancy in traditional leafy vegetables. The conclusion was edited as follows:

Conclusions

This review sheds light onto a better understanding of the regulation of seed dormancy and the main factors known to be involved in the control of seed dormancy at hormonal, transcriptomic, epigenetic, protein, and environmental levels. The abscisic–gibberellin acids (ABA-GA) balance is a key component involved in the indirect control of seed dormancy that determine whether seeds may germinate or not. Auxin and salicylic acid (SA) promote seed dormancy; jasmonic acid (JA), brassinosteroïds (BRs) and ethylene (ET) play a dual role in seed dormancy regulation, and cytokinin (CTKs) promotes seed germination. Other dormancy regulators were shown to directly promote or repress the decision of seeds to germinate. Those include genes such as Delay of Germination (DOG), DOF affecting germination (DAG), and reduced dormancy (RDO). For Gynandropsis gynandra and other TLVs, research is needed to solve the seed dormancy constraints and to provide prospective producers with high quality seeds. Gynandropsis gynandra, a closely related species to Arabidopsis thaliana, was considered as a model plant to propose a pathway into solving the dormancy constraints in TLVs. This proposed research avenue includes germplasm collection, germplasm characterization; development of mapping populations; QTL mapping; gene expression analysis; and new variety development. Therefore, researches are needed to highlight the specific storage conditions and seed pre-treatment required to ensure the seed viability and germination of genotypes. The mechanisms occurring during after-ripening in G. gynandra should also be elucidated. A rapid progress for G. gynandra full domestication should be achieved starting with the analysis of the natural variation in seed dormancy in the available genotype collections. Further steps include the characterization of the different factors influencing seed physiology, dormancy and dormancy breaking approaches. Also, the identification of major genetic and molecular factors underlying seed dormancy during seed development, seed storage and germination need to be elucidated for developing cultivars that farmers can effectively use. Multidisciplinary research teams including physiologists, geneticists and bioinformaticians are therefore required to quickly and efficiently make significant progress towards breeding for non-dormancy in traditional leafy vegetables.

Reviewer 3 Report

This article provided a detailed evidence of current knowledge base on molecular mechanism of seed dormancy in TLVs. The article is well written and easy to read. But I am concerned about the figures and table added. I would request the authors to consider the following points  before final revision.

Figure captions: I guess the figures were taken/modified from another source. It is important to provide credits to the original producer (s) of the figures. Providing a detail caption can be helpful for the reader. It seems to me that the figure captions became parts of text paragraph. I would suggest to separate the figure captions from the text. Tables: Tables 2 and 3 have overlapping information. I would suggest to provide only one table by merging both together. Figure 2. I would suggest to present figure2 upside down.

Author Response

Answers to comments, revisions and questions of Reviewer 3

Comment 23: This article provided a detailed evidence of current knowledge base on molecular mechanism of seed dormancy in TLVs. The article is well written and easy to read. But I am concerned about the figures and table added. I would request the authors to consider the following points before final revision.

Answer 23: We are thankful to the reviewer for this positive comment. The suggested revisions and comments have been used to improve the manuscript.

Comment 24: Figure captions: I guess the figures were taken/modified from another source. It is important to provide credits to the original producer (s) of the figures. Providing a detail caption can be helpful for the reader. It seems to me that the figure captions became parts of text paragraph. I would suggest to separate the figure captions from the text.

Answer 24: We designed the figure ourselves based on the information collected. This has not taken or modified from any another source. All sources where information are collected are cited in the Table 3.

Comment 25: Tables 2 and 3 have overlapping information. I would suggest to provide only one table by merging both together.

Answer 25: We suggest to keep both tables separate. Since Table 2 provides information on the expression of mutants of major genes studied in Arabidopsis species for seed dormancy release or germination inhibition. Whereas, Table 3 provides information necessary to understand clearly the Figure 1. It provides information about different processes involved in seed dormancy regulation and related species and genes.

Comment 26: Figure 2. I would suggest to present figure 2 upside down. 

Answer 26: Figure 2 has been designed accordingly in the revised manuscripts.